# Equivariant Deep Weight Space Alignment

## Abstract

Permutation symmetries of deep networks make simple operations like model averaging and similarity estimation challenging. In many cases, aligning the weights of the networks, i.e., finding optimal permutations between their weights, is necessary. More generally, weight alignment is essential for a wide range of applications, from model merging, through exploring the optimization landscape of deep neural networks, to defining meaningful distance functions between neural networks. Unfortunately, weight alignment is an NP-hard problem. Prior research has mainly focused on solving relaxed versions of the alignment problem, leading to either time-consuming methods or sub-optimal solutions. To accelerate the alignment process and improve its quality, we propose a novel framework aimed at learning to solve the weight alignment problem, which we name Deep-Align. To that end, we first demonstrate that weight alignment adheres to two fundamental symmetries and then, propose a deep architecture that respects these symmetries. Notably, our framework does not require any labeled data. We provide a theoretical analysis of our approach and evaluate Deep-Align on several types of network architectures and learning setups. Our experimental results indicate that a feed-forward pass with Deep-Align produces better or equivalent alignments compared to those produced by current optimization algorithms. Additionally, our alignments can be used as an initialization for other methods to gain even better solutions with a significant speedup in convergence.

## 1 Introduction

The space of deep network weights has a complex structure since networks maintain their function under certain permutations of their weights. This fact makes it hard to perform simple operations over deep networks, such as averaging their weights or estimating similarity. It is therefore highly desirable to "align" networks - find optimal permutations between the weight matrices of two networks. Weight Alignment is critical to many tasks that involve weight spaces. One key application is model merging and editing (Ainsworth et al., 2022; Wortsman et al., 2022; Stoica et al., 2023; Ilharco et al., 2022), in which the weights of two or more models are (linearly) combined into a single model to improve their performance or enhance their capabilities. Weight alignment algorithms are also vital to the study of the loss landscape of deep networks (Entezari et al., 2022), a recent research direction that has gained increasing attention. Moreover, weight alignment induces an invariant distance function on the weight space that can be used for clustering and visualization.

Since weight alignment is NP-hard (Ainsworth et al., 2022), current approaches rely primarily on local optimization of the alignment objective which is time-consuming and may lead to suboptimal solutions. Therefore, identifying methods with faster run time and improved alignment quality is an important research objective. A successful implementation of such methods would allow practitioners to perform weight alignment in real-time, for example, when merging models in federated or continual learning setups, or to perform operations that require computing many alignments in a reasonable time, such as weight space clustering.

Following a large body of works that suggested *learning* to solve combinatorial optimization problems using deep learning architectures (Khalil et al., 2017; Bengio et al., 2021; Cappart et al., 2021), we propose the first learning-based approach to weight alignment, called Deep-Align. Deep-Align is a neural network with a specialized architecture to predict high-quality weight alignments for a given distribution of data. A major benefit of our approach is that after a model has been trained, predicting the alignment between two networks amounts to a simple feed-forward pass through the

network followed by an efficient projection step, as opposed to solving an optimization problem in other methods.

This paper presents a principled approach to designing a deep architecture for the weight alignment problem. We first formulate the weight-alignment problem and prove it adheres to a specific equivariance structure. We then propose a neural architecture that respects this structure, based on newly suggested equivariant architectures for deep-weight spaces (Navon et al., 2023) called Deep Weight Space Networks (DWSNets). The architecture is based on a Siamese application of DWSNets to a pair of input networks, mapping the outputs to a lower dimensional space we call *activation space*, and then using a generalized outer product layer to generate candidates for optimal permutations.

Theoretically, we prove that our architecture can approximate the Activation Matching algorithm Tatro et al. (2020); Ainsworth et al. (2022), which computes the activations of the two networks on some pre-defined input data and aligns their weights by solving a sequence of linear assignment problems. This theoretical analysis suggests that DEEP-ALIGN can be seen as a learnable generalization of this algorithm. Furthermore, we show that DEEP-ALIGN has a valuable theoretical property called *Exactness*, which guarantees that it always outputs the correct alignment when there is a solution with zero objective.

Obtaining labeled training data is one of the greatest challenges when learning to solve combinatorial optimization problems. To address this challenge, we generate labeled examples on the fly by applying random permutations and noise to our unlabeled data. We then train our network using a combination of supervised and unsupervised loss functions *without* relying on any labeled examples.

Our experimental results indicate that DEEP-ALIGN produces better or comparable alignments relative to those produced by slower optimization-based algorithms, when applied to both MLPs and CNNs. Furthermore, we show that our alignments can be used as an initialization for other methods that result in even better alignments, as well as significant speedups in their convergence. Lastly, we show that our trained networks produce meaningful alignments even when applied to out-of-distribution weight space data.

**Previous work.** Several algorithms have been proposed for weight-alignment (Tatro et al., 2020; Ainsworth et al., 2022; Peña et al., 2023; Akash et al., 2022). Ainsworth et al. (2022) presented three algorithms: Activation Matching, Weight Matching, and straight-through estimation. Peña et al. (2023) improved upon these algorithms by incorporating a Sinkhorn-based projection method. In part, these works were motivated by studying the loss landscapes of deep neural networks. It was conjectured that deep networks exhibit a property called *linear mode connectivity*: for any two trained weight vectors (i.e., a concatenation of all the parameters of neural architecture), a linear interpolation between the first vector and the optimal alignment of the second, yields very small increases in the loss (Entezari et al., 2022; Garipov et al., 2018; Draxler et al., 2018; Freeman & Bruna, 2016; Tatro et al., 2020). Another relevant research direction is the growing area of research that focuses on applying neural networks to neural network weights. Early methods proposed using simple architectures (Unterthiner et al., 2020; Andreis et al., 2023; Eilertsen et al., 2020). Several recent papers exploit the symmetry structure of the weight space in their architectures (Navon et al., 2023; Zhou et al., 2023a;b; Zhang et al., 2023). A comprehensive survey of relevant previous work can be found in Appendix A.

## 2 PRELIMINARIES

**Equivariance** Let $G$ be a group acting on $\mathcal{V}$ and $\mathcal{W}$. We say that a function $L : \mathcal{V} \to \mathcal{W}$ is *equivariant* if $L(gv) = gL(v)$ for all $v \in \mathcal{V}, g \in G$.

**MultiLayer Perceptrons and weight spaces.** The following definition follow the notation in Navon et al. (2023). An $M$-layer MultiLayer Perceptron (MLP) $f_v$ is a parametric function of the following form:

$$f(x) = x_M, \quad x_{m+1} = \sigma(W_{m+1}x_m + b_{m+1}), \quad x_0 = x \tag{1}$$

Here, $x_m \in \mathbb{R}^{d_m}$, $W_m \in \mathbb{R}^{d_m \times d_{m-1}}$, $b_m \in \mathbb{R}^{d_m}$, and $\sigma$ is a pointwise activation function. Denote by $v = [W_m, b_m]_{m \in [M]}$ the concatenation of all (vectorized) weight matrices and bias vectors. We define the *weight-space* of an $M$-layer MLP as: $\mathcal{V} = \bigoplus_{m=1}^{M} (\mathcal{W}_m \oplus \mathcal{B}_m)$, where $\mathcal{W}_m :=$ $\mathbb{R}^{d_m \times d_{m-1}} \mathcal{B}_m = \mathbb{R}^{d_m}$ and $\bigoplus$ denotes the direct sum (concatenation) of vector spaces. A vector in

this space represents all the learnable parameters on an MLP. We define the *activation space* of an MLP as $\mathcal{A} = \bigoplus_{m=1}^{M} \mathbb{R}^{d_m} := \bigoplus_{m=1}^{M} \mathcal{A}_m$. The activation space, as its name implies, represents the concatenation of network activations at all layers. i.e., $\mathcal{A}_m$ is the space in which $x_m$ resides.

**Symmetries of weight spaces.** The permutation symmetries of the weight space are a result of the equivariance of pointwise activations: for every permutation matrix $P$ we have that $P\sigma(x) = \sigma(Px)$. Thus for example, a shallow network defined by weight matrices $W_1, W_2$ will represent the same function as the network defined by $PW_1, W_2 P^T$, since the permutations cancel each other. The same idea can be used to identify permutation symmetries of general MLPs of depth $M$. In this case, the weight space's symmetry group is the direct product of symmetric groups for each intermediate dimension $m \in [1, M-1]$ namely, $S_{d_1} \times \cdots \times S_{d_{M-1}}$. For clarity, we formally define the symmetry group as a product of matrix groups: $G = \Pi_{d_1} \times \cdots \times \Pi_{d_{M-1}}$, where $\Pi_d$ is the group of $d \times d$ permutation matrices (which is isomorphic to $S_d$). For $v \in \mathcal{V}$, $v = [W_m, b_m]_{m \in [M]}$, a group element $g = (P_1, \ldots, P_{M-1})$ acts on $v$ via a group action $v' = g_{\#}(v)$, where $v' = [W'_m, b'_m]_{m \in [M]}$ is defined by:

$$W'_1 = P_1 W_1, W'_M = W_M P_{M-1}^T, \text{ and } W'_m = P_m W_m P_{m-1}^T, \forall m \in [2, M-1]$$

$$b'_1 = P_1 b_1, \ b_{M'} = b_M, \text{ and } b'_m = P_m b_m, \forall m \in [2, M-1].$$

By construction, $v$ and $v' = g_{\#}(v)$ define the same function $f_v = f_{v'}$. The group product $g \cdot g'$ and group inverse $g^{-1} = g^T$ are naturally defined as the elementwise matrix product and transpose operations $g \cdot g' = (P_1 P'_1, \ldots, P_M P'_M)$, $g^T = (P_1^T, \ldots, P_m^T)$. Note that the elementwise product and transpose operations are well defined even if the $P_m$ and $P'_m$ matrices are not permutations.

## 3   THE WEIGHT ALIGNMENT PROBLEM AND ITS SYMMETRIES

**The weight alignment problem.** Given an MLP architecture as in equation 1 and two weight-space vectors $v, v' \in \mathcal{V}$, where $v = [W_m, b_m]_{m \in [M]}, v' = [W'_m, b'_m]_{m \in [M]}$, the weight alignment problem is defined as the following optimization problem:

$$\mathcal{G}(v, v') = \mathrm{argmin}_{k \in G} \|v - k_{\#} v'\|_2^2 \tag{2}$$

In other words, the problem seeks a sequence of permutations $k = (P_1, \ldots, P_{M-1})$ that will make $v'$ as close as possible to $v$. The optimization problem in equation 2 always admits a minimizer since $G$ is finite. For some $(v, v')$ it may have several minimizers, in which case $\mathcal{G}(v, v')$ is a set of elements. To simplify our discussion we will sometimes consider the domain of $\mathcal{G}$ to be only the set $\mathcal{V}_{\text{unique}}^2$ of pairs $(v, v')$ for which a unique minimizer exists. On this domain we can consider $\mathcal{G}$ as a function to the unique minimizer in $G$, that is $\mathcal{G} : \mathcal{V}_{\text{unique}}^2 \to G$.

Our goal in this paper is to devise an architecture that can learn the function $\mathcal{G}$. As a guiding principle for devising this architecture, we would like this function to be equivariant to the symmetries of $\mathcal{G}$. We describe these symmetries next.

**The symmetries of $\mathcal{G}$.** One important property of the function $\mathcal{G}$ is that it is equivariant to the action of the group $H = G \times G$ which consists of two independent copies of the permutation symmetry group for the MLP architecture we consider. Here, the action $h = (g, g') \in H$ on the input space $\mathcal{V} \times \mathcal{V}$ is simply $(v, v') \mapsto (g_{\#} v, g'_{\#} v')$, and the action of $h = (g, g') \in H$ on an element $k \in G$ in the output space is given by $g \cdot k \cdot g'^T$. This equivariance property is summarized and proved in the proposition below and visualized using the commutative diagram in Figure 1: applying $\mathcal{G}$ and then $(g, g')$ results in exactly the same output as applying $(g, g')$ and then $\mathcal{G}$.

**proposition 1.** *The map $\mathcal{G}$ is $H$-equivariant, namely, for all $(v, v') \in \mathcal{V}_{unique}^2$ and $(g, g') \in H$,*

$$\mathcal{G}(g_{\#} v, g'_{\#} v') = g \cdot \mathcal{G}(v, v') \cdot g'^T$$

The function $\mathcal{G}$ exhibits another interesting property: swapping the order of the inputs $v, v'$ corresponds to inverting the optimal alignment $\mathcal{G}(v, v')$:

**proposition 2.** *Let $(v, v') \in \mathcal{V}_{unique}^2$ then $\mathcal{G}(v', v) = \mathcal{G}(v, v')^T$.*

**Extension to multiple minimizers.** For simplicity the above discussion focused on the case where $(v, v') \in \mathcal{V}_{\text{unique}}^2$. We can also state analogous claims for the general case where multiple minimizers

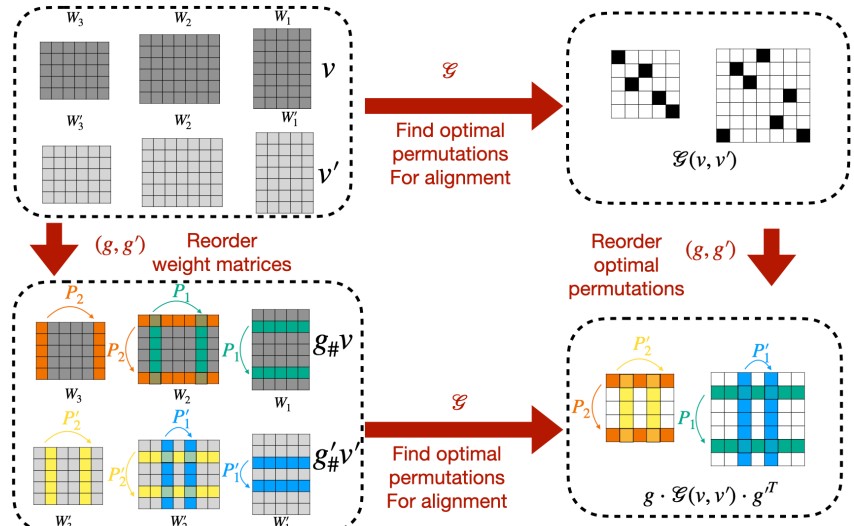

Figure 1: The equivariance structure of the alignment problem. The function $\mathcal{G}$ takes as input two weight space vectors $v, v'$ and outputs a sequence of permutation matrices that aligns them denoted $\mathcal{G}(v, v')$. In case we reorder the input using $(g, g')$ where $g = (P_1, P_2), g' = (P'_1, P'_2)$, the optimal alignment undergoes a transformation, namely $\mathcal{G}(g_\# v, g'_\# v') = g \cdot \mathcal{G}(v, v') \cdot g'^T$.

are possible. In this case we will have that the equalities $g \cdot \mathcal{G}(v, v') \cdot g'^T = \mathcal{G}(gv, g'v')$ and $\mathcal{G}(v, v')^T = \mathcal{G}(v', v)$ still hold as equalities between subsets of $G$.

**Extension to other optimization objectives.** In Appendix B we show that the equivariant structure of the function $\mathcal{G}$ occurs not only for the objective in equation 2, but also when the objective $\|v - k_\# v'\|_2^2$ is replaced with any scalar function $E(v, k_\# v')$ that satisfies the following properties: (1) $E$ is invariant to the action of $G$ on both inputs; and (2) $E$ is invariant to swapping its arguments.

## 4 DEEP-ALIGN

### 4.1 ARCHITECTURE

Here, we define a neural network architecture $F = F(v, v'; \theta)$ for learning the weight-alignment problem. The output of $F$ will be a sequence of square matrices $(P_1, \ldots, P_{M-1})$ that represents a (sometimes approximate) group element in $G$. In order to provide an effective inductive bias, we will ensure that our architecture meets both properties: 1,2, namely $F(g_\# v, g'_\# v') = g \cdot F(v, v') \cdot g'^T$ and $F(v, v') = F(v', v)^T$. The architecture we propose is composed of four functions:

$$F = F_{proj} \circ F_{prod} \circ F_{\mathcal{V} \to \mathcal{A}} \circ F_{DWS} : \mathcal{V} \times \mathcal{V}' \to \bigoplus_{m=1}^{M-1} \mathbb{R}^{d_m \times d_m},$$

where the equivariance properties we require are guaranteed by constructing each of the four functions composing $F$ to be equivariant with respect to an appropriate action of $H = G \times G$ and the transposition action $(v, v') \mapsto (v', v)$. In general terms, we choose $F_{DWS}$ to be a siamese weight space encoder, $F_{\mathcal{V} \to \mathcal{A}}$ is a siamese function that maps the weight space to the activation space, $F_{prod}$ is a function that performs (generalized) outer products between corresponding activation spaces in both networks and $F_{proj}$ performs a projection of the resulting square matrices on the set of doubly stochastic matrices (the convex hull of permutation matrices). The architecture is illustrated in Figure 2. We now describe our architecture in more detail.

**Weight space encoder .** $F_{DWS} : \mathcal{V} \times \mathcal{V}' \to \mathcal{V}^d \times \mathcal{V}'^d$, where $d$ represents the number of feature channels, is implemented as a Siamese DWSNet (Navon et al., 2023). This function outputs two weight-space embeddings in $\mathcal{V}^d$, namely, $F_{DWS}(v, v') = (\mathcal{E}(v), \mathcal{E}(v'))$, for a DWS network $\mathcal{E}$. The Siamese structure of the network guarantees equivariance to transposition. This

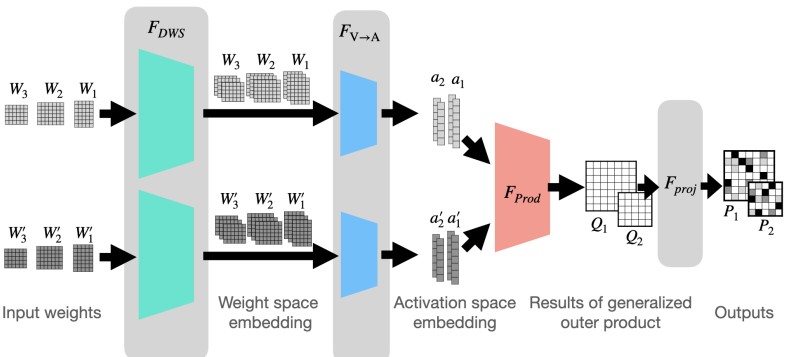

Figure 2: Our architecture is a composition of four blocks: The first block, $F_{DWS}$ generates weight space embedding for both inputs. The second block $F_{\mathcal{V} \to \mathcal{A}}$ maps these to the activation spaces. The third block, $F_{Prod}$, generates square matrices by applying an outer product between the activation vector of one network to the activation vectors of the other network. Lastly, the fourth block, $F_{Proj}$ projects these square matrices on the (convex hull of) permutation matrices.

is because the same encoder is used for both inputs, regardless of their input order. The $G$-equivariance of DWSNet, on the other hand, implies equivariance to the action of $G \times G$, that is $(\mathcal{E}(g_\# v), \mathcal{E}(g'_\# v')) = (g_\# \mathcal{E}(v), g'_\# \mathcal{E}(v'))$.

**Mapping the weight space to the activation space.** The function $F_{\mathcal{V} \to \mathcal{A}} : \mathcal{V}^d \times \mathcal{V}'^d \to \mathcal{A}^d \times \mathcal{A}'^d$ maps the weight spaces $\mathcal{V}^d, \mathcal{V}'^d$ to the corresponding Activation Spaces (see preliminaries section). There are several ways to implement $F_{\mathcal{V} \to \mathcal{A}}$. As the bias space, $\mathcal{B} = \bigoplus_{m=1}^{M} \mathcal{B}_m$, and the activation space have a natural correspondence between them, perhaps the simplest way, which we use in this paper, is to map a weight space vector $v = (w, b) \in \mathcal{V}^d$ to its bias component $b \in \mathcal{B}^d$. We emphasize that the bias representation is extracted from the previously mentioned weight space decodes, and in that case, it depends on and represents both the weights and the biases in the input . This operation is again equivariant to transposition and the action of $G \times G$, where the action of $G \times G$ on the input space is the more complicated action (by $(g_\#, g'_\#)$) on $\mathcal{V} \times \mathcal{V}$ and the action on the output space is the simpler action of $G \times G$ on the activation spaces.

**Generalized outer product.** $F_{prod} : \mathcal{A}^d \times \mathcal{A}'^d \to \bigoplus_{m=1}^{M} \mathbb{R}^{d_m \times d_m}$ is a function that takes the activation space features and performs a *generalized outer product* operation as defined below:

$$F_{prod}(a, a')_{m,i,j} = \phi([a_{m,i}, a'_{m,j}])$$

where the subscripts $m, i, j$ represent the $(i, j)$-th entry of the $m$-th matrix, and $a_{m,i}, a'_{m,j} \in \mathbb{R}^d$ are the rows of $a, a'$. Here, the function $\phi$ is a general (parametric or nonparametric) symmetric function in the sense that $\phi(a, b) = \phi(b, a)$. In this paper, we use $\phi(a, b) = s^2 \langle a/\|a\|_2, b/\|b\|_2 \rangle$ where $s$ is a trainable scalar scaling factor. The equivariance with respect to the action of $G \times G$ and transposition is guaranteed by the fact that $\phi$ is applied elementwise, and is symmetric, respectively.

**Projection layer.** The output of $F_{prod}$ is a sequence of matrices $Q_1, \ldots, Q_{M-1}$ which in general will not be permutation matrices. To bring the outputs closer to permutation matrices, $F_{proj}$ implements a approximate projection onto the convex hull of the permutation matrices, i.e., the space of doubly stochastic matrices. In this paper, we use two different projection operations, depending on whether the network is in training or inference mode. At training time, to ensure differentiability, we implement $F_{proj}$ as an approximation of a matrix-wise projection $Q_m$ to the space of doubly stochastic matrices using several iterations of the well-known Sinkhorn projection (Mena et al., 2018; Sinkhorn, 1967). Since the set of doubly stochastic matrices is closed under the action of $G \times G$ on the output space, and under matrix transposition, and since the Sinkhorn iterations are composed of elementwise, row-wise, or column-wise operations, we see that this operation is equivariant as well. At inference time, we obtain permutation matrices from $Q_i$ by finding the permutation matrix $P_i$ which has the highest correlation with $Q_i$, that is $P_i = \arg\max_{P \in S_{d_i}} \langle Q_i, P \rangle$,

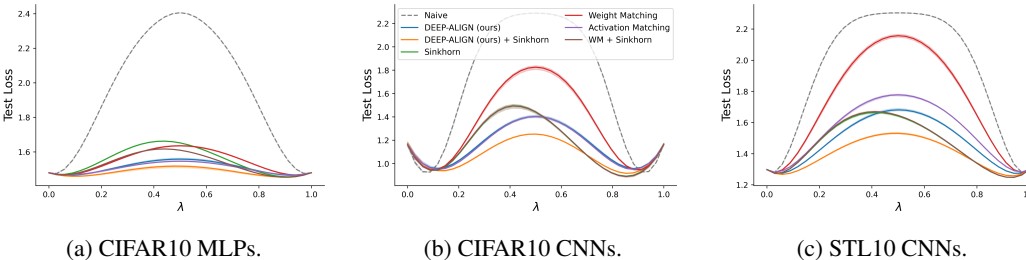

Figure 3: *Merging image classifiers*: the plots illustrate the values of the loss function used for training the input networks when evaluated on a line segment connecting $v$ and $g_{\#}v'$, where $g$ is the output of each method. Values are averaged over all test images and networks and 3 random seeds.

where the inner product is the standard Frobenious inner product. The optimization problem, known as the *linear assignment problem* can be solved using the Hungarian algorithm.

As we carefully designed the components of $F$ so that they are all equivariant to transposition and the action of $G \times G$, we obtain the following proposition:

**proposition 3.** *The architecture $F$ satisfies the conditions specified in 1,2, namely for all $(v, v') \in \mathcal{V} \times \mathcal{V}$ and $(g, g') \in H$ we have: $F(g_{\#}v, g'_{\#}v') = g \cdot F(v, v') \cdot g'^T$ and $F(v, v') = F(v', v)^T$.*

### 4.2 DATA GENERATION AND LOSS FUNCTIONS

Generating labeled data for the weight-alignment problem is hard due to the intractability of the problem. Therefore, we propose a combination of both unsupervised and supervised loss functions where we generate labeled examples synthetically from unlabeled examples, as specified below.

**Data generation.** Our initial training data consists of a finite set of weight space vectors $D \subset \mathcal{V}$. From that set, we generate two datasets consisting of pairs of weights for the alignment problem. First, we generate a labeled training set, $D_{\text{labeled}} = \{(v^j, v'^j, t^j)\}_{j=1}^{N_{\text{labeled}}}$ for $t^j = (T_1^j, \ldots, T_{M-1}^j) \in G$. This is done by sampling $v^j \in D$ and defining $v'^j$ as a permuted and noisy version of $v^j$. More formally, we sample a sequence of permutations $t \in G$ and define $v'^j = t_{\#}f_{\text{aug}}(v^j)$, where $f_{\text{aug}}$ applies several weight-space augmentations, like adding binary and Gaussian noise, scaling augmentations for ReLU networks, etc. We then set the label of this pair to be $t$. In addition, we define an unlabeled dataset $D_{\text{unlabeled}} = \{(v^j, v'^j)\}_{j=1}^{N_{\text{unlabeled}}}$ where $v^j, v'^j \in \mathcal{V}$.

**Loss functions.** The datasets above are used for training our architecture using the following loss functions. The labeled training examples in $D_{\text{labeled}}$ are used by applying a cross-entropy loss for each row $i = 1, \ldots, d_m$ in each output matrix $m = 1, \ldots, M - 1$. This loss is denoted as $\ell_{\text{supervised}}(F(v, v'; \theta), t)$. The unlabeled training examples are used in combination with two unsupervised loss functions. The first loss function aims to minimize the alignment loss in equation 2 directly by using the network output $F(v, v'; \theta)$ as the permutation sequence. This loss is denoted as $\ell_{\text{alignment}}(v, v', \theta) = \|v - F(v, v'; \theta)_{\#}v'\|_2^2$. The second unsupervised loss function aims to minimize the original loss function used to train the input networks on a line segment connecting the weights $v$ and the transformed version of $v'$ using the network output $F(v, v'; \theta)$ as the permutation sequence. Concretely, let $\mathcal{L}$ denote the original loss function for the weight vectors $v, v'$, the loss is defined as $\ell_{\text{LMC}}(v, v', \theta) = \mathcal{L}(\lambda v + (1 - \lambda)F(v, v'; \theta)_{\#}v')$ for $\lambda$ sampled uniformly $\lambda \sim U(0,1)$[1]. This loss is similar to the STE method in Ainsworth et al. (2022) and the differentiable version in Peña et al. (2023). Our final goal is to minimize the parameters of $F$ with respect to a linear (positive) combination of $\ell_{\text{alignment}}, \ell_{\text{LMC}}$ and $\ell_{\text{supervised}}$ applied to the appropriate datasets described above.

## 5 THEORETICAL ANALYSIS

**Relation to the activation matching algorithm.** In this subsection, we prove that our proposed architecture can simulate the activation matching algorithm, a heuristic for solving the weight align-

---

[1]This loss function satisfies the properties as described in Section 3 when taking expectation over $\lambda$.

Table 1: *MLP image classifiers*: Results on aligning MNIST and CIFAR10 MLP image classifiers.

| | MNIST (MLP) | | CIFAR10 (MLP) | |
|---|---|---|---|---|
| | Barrier ↓ | AUC ↓ | Barrier ↓ | AUC ↓ |
| Naive | $2.007 \pm 0.00$ | $0.835 \pm 0.00$ | $0.927 \pm 0.00$ | $0.493 \pm 0.00$ |
| Weight Matching | $0.047 \pm 0.00$ | $0.011 \pm 0.00$ | $0.156 \pm 0.00$ | $0.068 \pm 0.00$ |
| Activation Matching | $0.024 \pm 0.00$ | $0.007 \pm 0.00$ | $0.066 \pm 0.00$ | $0.024 \pm 0.00$ |
| Sinkhorn | $0.027 \pm 0.00$ | $0.002 \pm 0.00$ | $0.183 \pm 0.00$ | $0.072 \pm 0.00$ |
| WM + Sinkhorn | $0.012 \pm 0.00$ | $\mathbf{0.000 \pm 0.00}$ | $0.137 \pm 0.00$ | $0.050 \pm 0.00$ |
| DEEP-ALIGN | $0.005 \pm 0.00$ | $\mathbf{0.000 \pm 0.00}$ | $0.078 \pm 0.01$ | $0.029 \pm 0.00$ |
| DEEP-ALIGN + Sinkhorn | $\mathbf{0.000 \pm 0.00}$ | $\mathbf{0.000 \pm 0.00}$ | $\mathbf{0.037 \pm 0.00}$ | $\mathbf{0.004 \pm 0.00}$ |

Table 2: *CNN image classifiers*: Results on aligning CIFAR10 and STL10 CNN image classifiers.

| | CIFAR10 (CNN) | | STL10 (CNN) | | Runtime (Sec) ↓ |
|---|---|---|---|---|---|
| | Barrier ↓ | AUC ↓ | Barrier ↓ | AUC ↓ | |
| Naive | $1.124 \pm 0.01$ | $0.524 \pm 0.00$ | $1.006 \pm 0.00$ | $0.650 \pm 0.00$ | — |
| Weight Matching | $0.661 \pm 0.02$ | $0.178 \pm 0.01$ | $0.859 \pm 0.00$ | $0.453 \pm 0.00$ | $0.21$ |
| Activation Matching | $0.238 \pm 0.01$ | $\mathbf{0.000 \pm 0.00}$ | $0.479 \pm 0.00$ | $0.250 \pm 0.00$ | $7.52$ |
| Sinkhorn | $0.313 \pm 0.01$ | $\mathbf{0.000 \pm 0.00}$ | $0.366 \pm 0.00$ | $0.163 \pm 0.00$ | $79.81$ |
| WM + Sinkhorn | $0.333 \pm 0.01$ | $\mathbf{0.000 \pm 0.00}$ | $0.371 \pm 0.00$ | $0.165 \pm 0.00$ | $80.02 = 0.21 + 79.81$ |
| DEEP-ALIGN | $0.237 \pm 0.01$ | $\mathbf{0.000 \pm 0.00}$ | $0.382 \pm 0.01$ | $0.182 \pm 0.00$ | $0.44$ |
| DEEP-ALIGN + Sinkhorn | $\mathbf{0.081 \pm 0.00}$ | $\mathbf{0.000 \pm 0.00}$ | $\mathbf{0.232 \pm 0.00}$ | $\mathbf{0.097 \pm 0.00}$ | $80.25 = 0.44 + 79.81$ |

ment problem suggested in Ainsworth et al. (2022). In a nutshell, this algorithm works by evaluating two neural networks on a set of inputs and finding permutations that align their activations by solving a linear assignment problem using the outer product matrix of the activations as a cost matrix for every layer $m = 1, \ldots, M - 1$.

**proposition 4.** *(*DEEP-ALIGN *can simulate activation matching) For any compact set $K \subset \mathcal{V}$ and $x_1, \ldots, x_N \in \mathbb{R}^{d_0}$, there exists an instance of our architecture $F$ and weights $\theta$ such that for any $v, v' \in K$ for which the activation matching algorithm has a single optimal solution $g \in G$ and another minor assumption specified in the appendix, $F(v, v'; \theta)$ returns $g$.*

This result offers an interesting interpretation of our architecture: the architecture can simulate activation matching while optimizing the input vectors $x_1, \ldots, x_N$ as a part of their weights $\theta$.

**Exactness.** We now discuss the *exactness* of our algorithms. An alignment algorithm is said to be exact on some input $(v, v')$ if it can be proven to successfully return the correct minimizer $\mathcal{G}(v, v')$. For NP-hard alignment problems such as weight alignment, exactness can typically be obtained when restricting it to 'tame' inputs $(v, v')$. Examples of exactness results in the alignment literature can be found in Aflalo et al. (2015); Dym & Lipman (2017); Dym (2018). The following proposition shows that (up to probability zero events) when $v, v'$ are exactly related by some $g \in G$, our algorithm will retrieve $g$ *exactly*:

**proposition 5** (DEEP-ALIGN is exact for perfect alignments). *Let $F$ denote the* DEEP-ALIGN *architecture with non-constant analytic activations and $d \geq 2$ channels. Then, for Lebesgue almost every $v \in \mathcal{V}$ and parameter vector $\theta$, and for every $g \in G$, we have that $F(v, g_{\#}v, \theta) = g$.*

## 6 EXPERIMENTS

In this section, we evaluate DEEP-ALIGN on the task of aligning and merging neural networks. To support future research and the reproducibility of the results, we will make our source code and datasets publicly available upon publication.

**Evaluation metrics.** We use the standard evaluation metrics for measuring model merging (Ainsworth et al., 2022; Peña et al., 2023): Barrier and Area Under the Curve (AUC). For two inputs $v, v'$ the Barrier is defined by $\max_{\lambda \in [0,1]} \psi(\lambda) \equiv \mathcal{L}(\lambda v + (1-\lambda)v') - (\lambda \mathcal{L}(v) + (1-\lambda)\mathcal{L}(v'))$ where $\mathcal{L}$ denote the loss function on the original task. Similarly, the AUC is defined as the integral

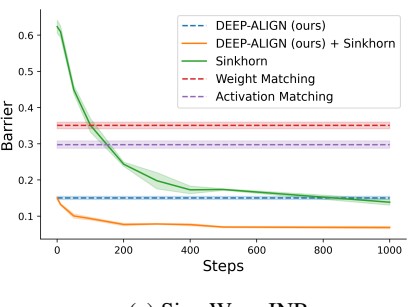 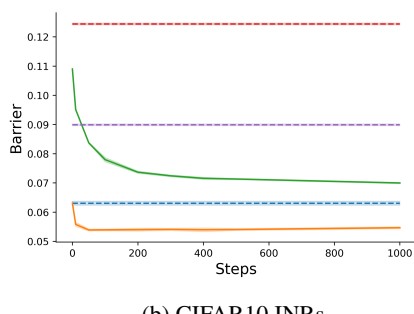

(a) Sine Wave INRs.

(b) CIFAR10 INRs.

Figure 4: *Aligning INRs*: The test barrier vs. the number of Sinkhorn iterations ( relevant only for *Sinkhorn* or DEEP-ALIGN + *Sinkhorn*), using (a) sine wave and (b) CIFAR10 INRs. DEEP-ALIGN outperforms baseline methods or achieves on-par results.

of $\psi$ over $[0, 1]$. Lower is better for both metrics. Following previous works (Ainsworth et al., 2022; Peña et al., 2023), we bound both metrics by taking the maximum between their value and zero.

**Compared methods.** We compare the following approaches: (1) *Naive*: where two models are merged by averaging the models' weights without alignment. The (2) *Weight matching* and (3) *Activation matching* approaches proposed in Ainsworth et al. (2022). (4) *Sinkhorn* (Peña et al., 2023): This approach directly optimizes the permutation matrices using the task loss on the line segment between the aligned models (denoted $\mathcal{C}_{Rnd}$ in Peña et al. (2023)). (5) *WM + Sinkhorn*: using the weight matching solution to initialize the Sinkhorn method. (6) DEEP-ALIGN: Our proposed method described in Section 4. (7) DEEP-ALIGN + *Sinkhorn*: Here, the output from the DEEP-ALIGN is used as an initialization for the *Sinkhorn* method.

**Experimental details.** Our method is first trained on a dataset of weight vectors and then applied to unseen weight vectors at test time, as is standard in learning setups. In contrast, baseline methods are directly optimized using the test networks. For the *Sinkhorn* and DEEP-ALIGN + *Sinkhorn* methods, we optimize the permutations for 1000 iterations. For the *Activation Matching* method, we calculate the activations using the entire train dataset. We repeat all experiments using 3 random seeds and report each metric's mean and standard deviation. For full experimental details see Appendix E.

### 6.1 RESULTS

**Aligning classifiers.** Here, we evaluate our method on the task of aligning image classifiers. We use four network datasets. Two datasets consist of MLP classifiers for MNIST and CIFAR10, and two datasets consist of CNN classifiers trained using CIFAR10 and STL10. This collection forms a diverse benchmark for aligning NN classifiers. The results are presented in Figure 7, Table 1 and Table 2. The alignment produced through a feed-forward pass with DEEP-ALIGN performs on par or outperforms all baseline methods. Initializing the Sinkhorn algorithm with our alignment (DEEP-ALIGN + Sinkhorn) further improves the results, and significantly outperforms all other methods. For the CNN alignment experiments, we report the averaged alignment time using 1K random pairs.

**Aligning INRs.** We use two datasets consisting of implicit neural representations (INRs). The first consists of Sine waves INRs of the form $f(x) = \sin(ax)$ on $[-\pi, \pi]$, where $a \sim U(0.5, 10)$, similarly to the data used in Navon et al. (2023). We fit two views (independently trained weight vectors) for each value of $a$ starting from different random initializations and the task is to align and merge the two INRs. We train our network to align pairs of corresponding views. The second dataset consists of INRs fitted to CIFAR10 images. We fit five views per image. The results are presented in Figure 4. DEEP-ALIGN, performs on par or outperforms all baseline methods. Moreover, using the output from the DEEP-ALIGN to initialize the Sinkhorn algorithm further improves this result, with a large improvement over the Sinkhorn baseline with random initialization.

**Generalization to out-of-distribution data (OOD).** Here, we evaluate the generalization capabilities of DEEP-ALIGN under distribution shift. We use the DEEP-ALIGN model trained on CIFAR10 CNN image classifiers and evaluate the generalization on two datasets. The first dataset consists of CNN classifiers trained on a version of CIFAR10 in which each image is rotated by a rotation

Table 3: Aligning OOD image classifiers, using a DEEP-ALIGN network trained on CIFAR10.

| | Rotated CIFAR10 | | STL10 | |
|---|---|---|---|---|
| | Barrier ↓ | AUC ↓ | Barrier ↓ | AUC ↓ |
| Naive | $1.077 \pm 0.01$ | $0.714 \pm 0.00$ | $1.006 \pm 0.00$ | $0.650 \pm 0.00$ |
| Weight Matching | $0.945 \pm 0.02$ | $0.550 \pm 0.01$ | $0.859 \pm 0.00$ | $0.453 \pm 0.00$ |
| Activation Matching | $0.586 \pm 0.00$ | $0.336 \pm 0.00$ | $0.479 \pm 0.00$ | $0.250 \pm 0.00$ |
| Sinkhorn | $0.596 \pm 0.01$ | $0.321 \pm 0.00$ | $0.366 \pm 0.00$ | $\mathbf{0.163 \pm 0.00}$ |
| DEEP-ALIGN | $0.769 \pm 0.01$ | $0.453 \pm 0.00$ | $0.686 \pm 0.01$ | $0.373 \pm 0.01$ |
| DEEP-ALIGN + Sinkhorn | $\mathbf{0.430 \pm 0.01}$ | $\mathbf{0.245 \pm 0.00}$ | $\mathbf{0.357 \pm 0.00}$ | $0.165 \pm 0.00$ |

degree sampled uniformly from $U(-45, 45)$. The second dataset consists of CNN image classifiers trained on the STL10 dataset. Importantly, we note that DEEP-ALIGN is evaluated on a distribution of models that is different than the one observed during training. In contrast, the baselines directly solve an optimization problem for each model pair within the test datasets. While DEEP-ALIGN significantly outperforms the Naive and WM baselines, it falls short in comparison to the Sinkhorn and AM methods, both of which are directly optimized using data from the new domain (NNs and images). Employing DEEP-ALIGN as an initialization for the Sinkhorn method consistently proves beneficial, with the DEEP-ALIGN + Sinkhorn approach yielding the most favorable results.

**Aligning networks trained on disjoint datasets.**
Following Ainsworth et al. (2022), we experiment with aligning networks trained on disjoint datasets. One major motivation for such a setup is Federated learning (McMahan et al., 2017). In Federated Learning, the goal is to construct a unified model from multiple networks trained on separate and distinct datasets.

To that end, we split the CIFAR10 dataset into two splits. The first consists of $95\%$ images from classes 0-4 and $5\%$ of classes 5-9, and the second split is constructed accordingly with $95\%$ of classes 5-9. We train the DEEP-ALIGN model to align CNN networks trained using the different datasets. For Sinkhorn and Activation Matching, we assume full access to the training data in the

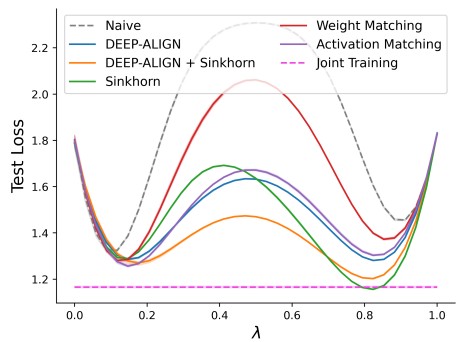

Figure 5: Merging networks trained on distinct subsets of CIFAR10.

optimization stage. For DEEP-ALIGN, we assume this data is accessible in the training phase. The results are presented in Figure 5. DEEP-ALIGN, along with the Sinkhorn and Activation Matching approaches, are able to align and merge the networks to obtain a network with lower loss compared to the original models. However, our approach is significantly more efficient at inference.

## 7 CONCLUSION

We investigate the challenging problem of weight alignment in deep neural networks. The key to our approach, DEEP-ALIGN, is an equivariant architecture that respects the natural symmetries of the problem. At inference time DEEP-ALIGN can align unseen network pairs without the need for performing expensive optimization. DEEP-ALIGN, performs on par or outperforms optimization-based approaches while significantly reducing the runtime or improving the quality of the alignments. Furthermore, we demonstrate that the alignments of our method can be used to initialize optimization-based approaches. One limitation of our approach is the need for training a network. Although this can be a relatively time-consuming process, we only have to perform it once for each weight distribution. Furthermore, this procedure does not require labeled examples. To summarize, DEEP-ALIGN is the first architecture designed for weight alignment. It demonstrates superior performance over existing methods. The generalization capabilities of DEEP-ALIGN make it a promising and practical solution for applications that require weight alignment.

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

## A  MORE PREVIOUS WORK

**Weight space alignment.** Several algorithms have been proposed for weight-alignment (Tatro et al., 2020; Ainsworth et al., 2022; Peña et al., 2023; Akash et al., 2022). Ainsworth et al. (2022) presented three algorithms: Activation Matching, Weight Matching, and straight-through estimation. Peña et al. (2023) improved upon these algorithms by incorporating a Sinkhorn-based projection method. In part, these works were motivated by studying the loss landscapes of deep neural networks. It was conjectured that deep networks exhibit a property called *linear mode connectivity*: for any two trained weight vectors, a linear interpolation between the first vector and the optimal alignment of the second, yields very small increases in the loss (Entezari et al., 2022; Garipov et al., 2018; Draxler et al., 2018; Freeman & Bruna, 2016; Tatro et al., 2020; Jordan et al., 2022).

**Weight-space networks.** A growing area of research focuses on applying neural networks to neural network weights. Early methods proposed using simple architectures such as MLPs or transformers to predict test errors or the hyperparameters that were used for training input networks (Unterthiner et al., 2020; Andreis et al., 2023; Eilertsen et al., 2020).

Recently, Navon et al. (2023) presented the first neural architecture that is equivariant to the natural permutation symmetries of (MLP) weight spaces and demonstrated significant performance improvements over previous approaches. This architecture, called Deep Weight Space Networks (DWSNets), is composed of multiple linear $G$-equivariant layers, which were characterized in Navon et al. (2023), interleaved with pointwise nonlinearities, such as ReLU functions. In other words, a DWSNets is a function $F : \mathcal{V} \to \mathcal{V}$ of the following form:

$$F = L_k \circ \sigma \circ L_{k-1} \circ \sigma \cdots \circ \sigma \circ L_1,$$

where $L_i$ are linear $G$-equivariant layers and $\sigma : \mathbb{R} \to \mathbb{R}$ is a nonlinear function applied elementwise. This is similar to the design of many previous equivariant architectures (Zaheer et al., 2017; Maron et al., 2018). Each such layer $L : \mathcal{V} \to \mathcal{V}$ takes as input representations of all the weights and biases which we denote as $v \in \mathcal{V}$, $v = [W_m, b_m]_{m \in [M]}$ and outputs new representations based on all the input weights and biases. As common in deep learning architectures, these layers can also handle $d$ dimensional features for each weight and bias, i.e. vectors $v \in \mathcal{V}^d$. After a composition of several such layers, the output weights can be used for the task of interest, or pooled to form a single representation for the input weight space vector. In our case, we use the bias representations produced by the final layer $L_k$, which are essentially representations of the activation space $\mathcal{A}$.

Zhou et al. (2023a) proposed a similar approach and an extension to CNN architectures , which was later enhanced by the addition of attention mechanisms (Zhou et al., 2023b). Finally, Zhang et al. (2023) proposes modeling the neural networks as computational graphs and applying Graph Neural Networks to them, demonstrating very good results on several weight space learning tasks.

**Learning for combinatorial optimization.** There exists a large body of research on learning to solve hard combinatorial optimization problems such as TSP and SAT (Cappart et al., 2021; Bengio et al., 2021; Khalil et al., 2017; Vesselinova et al., 2020; Selsam et al., 2018). The key observation behind these works, also shared by the current work, is that even though those problems are computationally intractable in general, there may be efficient methods to solve them for specific problem distributions. In these cases, machine learning can be used to learn to predict these solutions. Specifically relevant to this work are works that suggested learning to solve the graph matching problem Fey et al. (2020); Zanfir & Sminchisescu (2018); Yan et al. (2020); Yu et al. (2019); Nowak et al. (2017; 2018) which is a similar alignment problem.

## B  PROOFS FOR SECTION 3

*Proof of Proposition 1.* We write $\mathcal{G}(v, v') = \text{argmin}_{k \in G} E(k, v, v')$ with $E(k, v, v') = \|v - k_\# v'\|_2^2$. First, we note that the minimal value of the optimization problem $\mathcal{G}(gv, gv')$ is equal to the minimal value of $\mathcal{G}(v, v')$, namely,

$$\min_{k \in G} E(k, v, v') = \min_{k \in G} E(k, g_\# v, g'_\# v').$$

This is true since

$$\min_{k \in G} E(k, g_\# v, g'_\# v') = \min_{k \in G} \|g_\# v - k_\# g'_\# v'\|_2^2 = \min_{k \in G} \|v - (g^{-1}kg')_\# v'\|_2^2 = \min_{k \in G} \|k_\# v - v'\|_2^2 =$$

and using the fact that $k \mapsto g^{-1}kg'$ is bijective.

Second, we show that if $k^* = \mathcal{G}(v, v')$, or in other words, is a minimizer of $E(k, v, v')$, then $g \cdot k^* \cdot g'^T$ minimizes $E(k, gv, g'v')$ because:

$$E(g \cdot k^* \cdot g'^T, gv, g'v') = \|g_\# v - (g \cdot k^* \cdot g'^T)_\# g'_\# v'\|_2^2 = \|v - (g^T \cdot g \cdot k^* \cdot g'^T \cdot g')_\# v'\|^2 = E(k^*, v, v')$$

$\square$

*Proof of Proposition 2.* Similarly to the proof of the previous proposition, we have $\min_{k \in G} E(k, v', v) = \min_{k \in G} \|v' - k_\# v\|_2^2 = \min_{k \in G} \|k_\#^T v' - v\|_2^2 = \min_{k \in G} E(k, v, v')$. Additionally, plugging in $k = \mathcal{G}(v, v')^T$ achieves this value:

$$E(\mathcal{G}(v, v')^T, v', v) = \|v' - [\mathcal{G}(v, v')^T]_\# v\|_2^2 = \|[\mathcal{G}(v, v')]_\# v' - v\|_2^2 = E(\mathcal{G}(v, v'), v, v')$$

$\square$

*Proof of generalization to other objectives.* We prove the generalization mentioned in the main text. Namely, that the symmetries of the function $\mathcal{G}$, are shared by any function of the form $\mathcal{G}(v, v') = \operatorname{argmin}_{k \in G} E(v, k_\# v')$ providing that the energy function $E$ satisfies

$$E(gv, gv') = E(v, v') \text{ and } E(v, v') = E(v', v), \quad \forall g \in G, v, v' \in \mathcal{V}.$$

First, we note that the minimal value of the optimization problem $\mathcal{G}(gv, gv')$ is equal to the minimal value of $\mathcal{G}(v, v')$, namely,

$$\min_{k \in G} E(v, k_\# v') = \min_{k \in G} E(g_\# v, k_\# g'_\# v').$$

This is true since

$$\min_{k \in G} E(g_\# v, k_\# g'_\# v') = \min_{k \in G} E(v, g_\#^T k_\# g'_\# v') = \min_{k \in G} E(v, (g^{-1}kg')_\# v')$$

using the fact that $k \mapsto g^{-1}kg'$ is bijective and the $G$-invariance of $E$.

Next, we show that if $k^* = \mathcal{G}(v, v')$, or in other words, is a minimizer of $E(v, k_\# v')$, then $g \cdot k^* \cdot g'^T$ minimizes $E(g_\# v, k_\# g'_\# v')$ This is because: $E(g_\# v, (g \cdot k^* \cdot g'^T)_\# g'_\# v') = E(v, k_\#^* v')$ using the $G$-invariance of $E$.

We turn to proving a generalization of 2: similarly to the previous argument, we have

$$\min_{k \in G} E(v', k_\# v) = \min_{k \in G} E(k_\#^T v', v) = \min_{k \in G} E(k_\#^T v', v) = \min_{k \in G} E(v, k_\#^T v')$$
$$= \min_{k \in G} E(v, k_\# v')$$

where we used the fact that we can swap the inputs to $E$ and the invariance.

Additionally, plugging in $k = \mathcal{G}(v, v')^T$ achieves this value:

$$E(v', \mathcal{G}(v, v')_\#^T v) = E(\mathcal{G}(v, v')_\# v', v) = E(v, \mathcal{G}(v, v')_\# v')$$

$\square$

## C  PROOFS FOR SECTION 5

**Relation to activation matching.** Here we prove Proposition 4. Let the outer product matrix $Z \in \mathbb{R}^{d_m \times d_m}$ be a cost matrix for the activation matching algorithm. We say that Z is $\epsilon$-*friendly* if it has the following property: the exists some $i$ such that for all $j \neq i$ $\langle Z, P_i \rangle > \langle Z, P_j \rangle + \epsilon$ where $P_i, P_j \in \Pi_{d_m}$. Intuitively this condition means that the cost matrix is bounded away from the set of cost matrices for which the are multiple optimal solutions. Let us now state Proposition 4 with full details:

**proposition 6** (Full formulation of Proposition 4). *For any compact set $K \subset \mathcal{V}$, $x_1, \ldots, x_N \in \mathbb{R}^{d_0}$ and for any $\epsilon > 0$, there exists an instance of our architecture $F$ and weights $\theta$ for that architecture such that for any $v, v' \in K$ for which all the cost matrices used by the activation matching algorithm are $\epsilon$-friendly, we have that $F(v, v'; \theta)$ returns exactly the same solution as the activation matching algorithm.*

*Proof of Proposition 6.* First, to obtain weights for the DWS network that approximate the activations of both networks on the set of inputs $x_1, \ldots, x_n$, we employ Lemma G.2 from Navon et al. (2023) (stated below for convenience). We note that this lemma is proved by the approximation of all intermediate activations of the input networks so the proof of this lemma trivially shows that there is a DWSnet that outputs an approximation of all the activations.

Then, we set $F_{prod}$ to calculate outer products in order to generate the outer-product-based cost matrix used in Ainsworth et al. (2022). It follows that the composition of the DWS network above with the outer product function approximates uniformly their limits, see Lim et al. (2022) (Lemma 6), and the composition of their limits is exactly the function that takes two weight vectors and computes the cost matrices from Ainsworth et al. (2022). We have developed an architecture and weights for this architecture that can uniformly approximate the cost matrices used by the activation matching algorithm to any precision. As a final step, we apply our linear assignment projection.

To show that this architecture will return exactly the same solution $g$ of the activation matching algorithm, we use our $\epsilon$-friendly assumption and Lemma 8 which imply that there is an open ball of radius $\delta$ around each cost matrix in which the linear assignment problem is *constant*. We can now use the uniform approximation result from the previous paragraph to approximate all the cost matrices up to $\delta$ and get that applying the linear assignment problem to the approximated cost matrices yields the same output as the original cost matrices.

$\square$

**Lemma 7.** *[Lemma G.2 in Navon et al. (2023)] Let $\epsilon > 0$. For any $x_1, \ldots, x_N \in \mathbb{R}^{d_0}$ there exist a DWSNet $D_\epsilon$ and weights $\theta_\epsilon$ such that for any $v \in K$ we have $\|D(v; \theta_\epsilon)_i - f_v(x_i)\|_2 < \epsilon$.*

We note that Navon et al. (2023) restricted $x_1, \ldots, x_N$ to some compact set but this assumption is not used in their proof.

**Lemma 8.** *Let $\epsilon > 0$, $K \subset \mathbb{R}^n$ a compact domain and let $f_i : K \to \mathbb{R}$, $i = 1, \ldots, m$ such that $f_i(x)$ are continuous. Let*

$$S = \{x \in K \mid \exists i \ s.t \ \forall j \neq i \ f_i(x) > f_j(x) + \epsilon\},$$

*define $g(x) = argmax_{j=1}^m f_j(x)$, then there exists some $\delta > 0$ such that for any $x \in S$, $g(x)$ is constant in an open ball of radius $\delta$ around $x$.*

*Proof.* Since $f_1, \ldots, f_n$ are finite and continuous on a compact domain, there exists some $\delta > 0$ such that for any $x, y \in K$, $\|x - y\| < \delta$ implies $|f_i(x) - f_i(y)| < \frac{\epsilon}{2}$ for every $i$. Then for any $x \in S$ such that $\forall j \neq i \ f_i(x) > f_j(x) + \epsilon$, if $\|x - y\| < \delta$, for every $j \neq i$ we have

$$f_i(y) - f_j(y) = (f_i(y) - f_i(x)) + (f_i(x) - f_j(x)) + (f_j(x) - f_j(y)) > 0,$$

which implies that $g(y) = i$. $\square$

*Proof of Proposition 5.* As mentioned in the statement of the proposition, we assume that $F$ uses analytic non-constant activation functions, and $d \geq 2$ channels. In the proof we consider our standard choice of $\phi$ as the normalized inner product and set the scaling parameter $s$ to be $s = 1$ for simplicity. We consider the version of $F$ where the last projection layer $F_{proj}$ computes the permutations maximizing the correlation with the outputs $Q_1, \ldots, Q_M$ of the product layer (that is, we consider the version of $F$ used in test time, rather than the version used in training which uses differentiable Sinkhorn iterations). By equivariance of the model, it is sufficient to show that for almost every $v, \theta$ the identity matrices are the closest permutations to $(Q_1, \ldots, Q_{M-1})$.

Next, recall that the indices of each $Q_m = Q_m(v, \theta)$ are given by

$$Q_{m,i,j} = \frac{\langle a_{m,i}, a_{m,j} \rangle}{\|a_{m,j}\| \|a_{m,j}\|}$$

where $a_{m,i}$ is a $d$ dimensional vector. In particular, we have that $|Q_{m,i,j}| \leq 1$, and the equality holds if $i = j$. It remains to show that when $i \neq j$ we will have a strict inequality $|Q_{m,i,j}| < 1$ for all $m$, for almost every $v$ and $\theta$. Equivalently, we will need to show that for all $m = 1, \ldots, M$ and all $i \neq j$, the functions

$$\phi_{m,i}(v, \theta) = \|a_{m,i}\|^2$$

and

$$\psi_{m,i,j}(v, \theta) = \|a_{m,j}\|^2 + \|a_{m,i}\|^2 - \langle a_{m,i}, a_{m,j} \rangle$$

are non-zero for almost all $(v, \theta)$. We note that $\phi$ and $\psi$ are analytic, and the zero set of a non-zero analytic function always has Lebesgue measure zero (see Mityagin (2015)). Therefore it is sufficient to show that there exists a single $(v, \theta)$ for which $\phi_{m,i}(v, \theta) \neq 0$, and a single $(v, \theta)$ for which $\psi_{m,i,j}(v, \theta) \neq 0$.

Let us consider parameter vectors $\theta$ as follows: recall that the output of each layer is a sequence of hidden weight matrices $W_m = W_{m,a,b,c}$ and a sequence of hidden bias vectors $b_m = b_{m,i,c}$, where the last index $c$ runs over the channels. We choose $\theta$ so that each affine layer will map all matrices $W_m$ to zero, and all bias vectors $b_m$ to a new value $b'_m$, where $b'_{m,i,1} = b_{m,i,1}$ and $b'_{m,i,c} = 1$ if $c \neq 1$. Since this describes an affine equivariant mapping, and the linear layers in DWS can express all linear equivariant function, the vector $\theta$ can indeed be defined to give this function.

With this choice of $\theta$, we will obtain

$$a_{m,i} = \left[ \rho^D(b_{m,i}), 1_{d-1} \right] \in \mathbb{R}^d$$

where $\rho$ denotes the activation used in the DWS network, $D$ denotes the depth of the DWS network, and $b_{m,i}$ is the $i$-th entry of the $m$-th input bias vector. In particular, it is an entry of $v$. To conclude the proof it is sufficient to show that we can choose a pair of $b_i, b_j$ such that $\rho^D(b_{m,i}) \neq \rho^D(b_{m,j})$. If this is indeed the case we can immediately deduce that $\psi_{m,i,j}(v, \theta) \neq 0$.

To prove the latter point all we need is to show that $\rho^D$ is not a constant function. Indeed, since $\rho$ itself is non-constant its image contains an interval $I$, and by analyticity $\rho$ cannot be constant on this interval, so $\rho^2$ is non-constant. Continuing recursively in this way we can show that $\rho^D$ will not be constant for any $D$. $\qquad\square$

## D  EXTENDING DWSNETS TO CNNS

In this work, we employed a DWSNet (Navon et al., 2023) as our $F_{DWS}$ block. Since the study in Navon et al. (2023) focused on MLPs, the original implementation only supports input MLP networks. Here, we provide technical details on the extension to input CNN networks under the DWSNets framework. This requires only two simple adjustments. First, the kernel dimensions are flattened into the feature dimension. Second, the first FC layer (after the last convolution layer) in the input network generally requires special attention. Specifically, denoting the output dimension of the last CNN with $d_0$ and the dimensions of the first FC layer weight matrix by $d_1 \times d_2$, we reshape the weight matrix to $d_0 \times d_2 \times (d_1/d_0)$, i.e., folding the $d_1/d_0$ into the feature dimension. This preserves the equivariance to the permutation symmetries of the input network.

## E  EXPERIMENTAL DETAILS

In all experiments, we use a 4-hidden layer DEEP-ALIGN network with a hidden dimension of 64 and an output dimension of 128 from the $F_{DWS}$ block. We optimize our method with a learning rate of $5e - 4$ using the AdamW (Loshchilov & Hutter, 2017) optimizer. For all experiments with image classifiers, we train DEEP-ALIGN with all objectives as described in Section 4 ($\ell_{\text{alignment}}, \ell_{\text{LMC}}$ and $\ell_{\text{supervised}}$). For the INR experiments, we drop the $\ell_{\text{alignment}}$ loss since we found adding this loss significantly hurt the performance (see Appendix F).

We use the entire dataset to estimate the activations for the Activation Matching (AM) baseline. For the Sinkhorn baseline, we optimize the permutations using learning rate $1e - 1$ and for 1000 iterations.

When using image datasets, we use the standard train-test split and allocate 10% of the training data for validation.

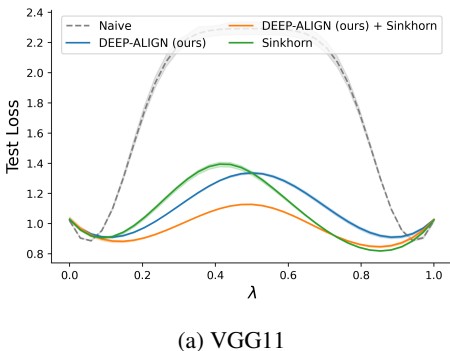 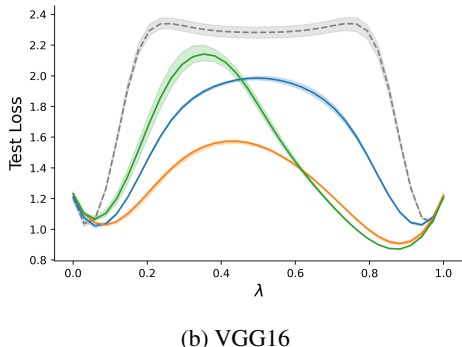

(a) VGG11                           (b) VGG16

Figure 6: Results for aligning VGG models trained using the CIFAR10 dataset. DEEP-ALIGN scales well to large networks (15M parameters for VGG16) without the need for increasing the training data size.

**MLP classifiers.** For this experiment, we generate two wight datasets, consisting of MNIST and CIFAR10 classifiers. Each classifier is a 3-hidden layer MLP with a hidden dimension of $128$. The input dimension is $784$ for MNIST and $3072$ for CIFAR10. We train the classifiers for $5$ epochs with a batch size of $128$ and learning rate $5e-3$. Both datasets consist of $10000$ networks, split into $8000$ for training and $1000$ each for validation and testing.

We train DEEP-ALIGN for 25K iterations. Since the $F_{DWS}$ block can grow large when the input dimension to the input network is large, we employ the method proposed in Navon et al. (2023) to control the number of parameters. Thus, we linearly map the input dimension $784$ or $3072$ (MNIST or CIFAR10) to $8$. See Navon et al. (2023) for details.

**CNN classifiers.** For this experiment, we generate two datasets by training classifiers on CIFAR10 and STL10 datasets. Each network consists of 7 CNN layers followed by a fully-connected layer. The full architecture is presented in Table 4. For STL10, we apply a sequence of 3 augmentations, first, we random crop $64 \times 64$ path, next we resize the patch to $32 \times 32$, and finally we apply random rotation drawn from $U(-20, 20)$. We train the CIFAR10 and STL10 classifiers for 20/100 epochs respectively using Adam optimizer with $1e-4$ learning rate. We save the model's checkpoint at the final epoch. Both datasets consist of $5000$ networks, split into $4500$ for training and $250$ each for validation and testing. We train DEEP-ALIGN for 300 epochs.

Table 4: CNN classifiers architecture.

| CNN Classifiers Arch. |
| --- |
| 3x3 Conv 16 |
| 3x3 Conv 32 |
| 3x3 Conv 32 |
| 2x2 MaxPool |
| 3x3 Conv 64 |
| 3x3 Conv 64 |
| 2x2 MaxPool |
| 3x3 Conv 128 |
| 3x3 Conv 128 |
| 2x2 MaxPool |
| Linear (2048, 10) |

**Sine INRs.** To generate the Sine wave dataset, we use the same procedure as in Navon et al. (2023). Each INR is an MLP with 3 layers, a hidden dimension of 32, and Sine activations. The dataset consists of 2000 sine waves and two INR copies (views) for each sine wave. We use 1800 waves for training, and 100 for validation, and testing.

**CIFAR10 INRs.** The CIFAR10 dataset consists of $60K$ images. We split the dataset to train, validation, and test with $45K$ / $5K$ / $10K$ samples respectively. For each image, we create $5$ independent INR copies. Each INR is a 5-layer MLP with a 32 hidden dimension each followed by sine activations. We optimize the INRs using Adam optimizer with a $1e-4$ learning rate for $10K$ update steps. We train DEEP-ALIGN for 300 epochs.

**Generalization.** To evaluate the generalization to STL10, we use the same STL10 CNN dataset from Section 6.1. Furthermore, we generate an additional test dataset of 100 CIFAR10 classifiers trained on images with random rotations.

**Disjoint datasets.** We use the same CNN network configuration as in the CNN classifiers experiments, and train 2500 networks for each split (5000 in total). Each network is trained for 20 epochs using the Adam optimizer with learning rate $1e-4$. We allocate 100 networks from each split for

Table 5: Scaling to large NNs: Results for aligning VGG11 (9M parameters) and VGG16 (15M parameters) networks trained on the CIFAR10 dataset.

| | CIFAR10 VGG11 | | CIFAR10 VGG16 | | Runtime (Sec) ↓ |
|---|---|---|---|---|---|
| | Barrier ↓ | AUC ↓ | Barrier ↓ | AUC ↓ | |
| Naive | $1.273 \pm 0.04$ | $0.738 \pm 0.02$ | $1.131 \pm 0.04$ | $0.777 \pm 0.03$ | − |
| Sinkhorn | $0.368 \pm 0.00$ | $0.041 \pm 0.00$ | $0.921 \pm 0.05$ | $0.212 \pm 0.02$ | 158.40 |
| DEEP-ALIGN | $0.310 \pm 0.00$ | $0.059 \pm 0.00$ | $0.779 \pm 0.01$ | $0.356 \pm 0.01$ | 0.98 |
| DEEP-ALIGN + Sinkhorn | $\mathbf{0.099 \pm 0.01}$ | $\mathbf{0.000 \pm 0.00}$ | $\mathbf{0.349 \pm 0.01}$ | $\mathbf{0.027 \pm 0.01}$ | $159.38 = 0.98 + 158.40$ |

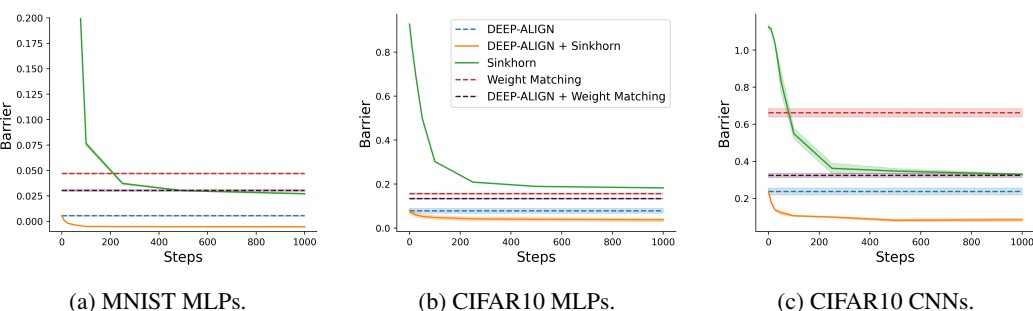

(a) MNIST MLPs.      (b) CIFAR10 MLPs.      (c) CIFAR10 CNNs.

Figure 7: DEEP-ALIGN as initialization: Results for using DEEP-ALIGN as initialization for the optimization-based approaches Sinkhorn re-basin and weight matching.

testing and validation, and the remaining 4600 networks for training. We train DEEP-ALIGN for 50K steps.

**Time comparison.** Prior methods for weight matching, which rely on optimization, often suffer from exhaustive runtime which may be impractical for real-time applications. In contrast, once trained, DEEP-ALIGN is able to produce high-quality weight alignments through a single forward pass and an efficient projection step. We compare DEEP-ALIGN to baselines by measuring the time required to align a pair of models in the CIFAR10 CNN classifiers dataset, and report the averaged alignment time using 1000 random pairs on a single RTX 2080-Ti Nvidia GPU. The results are presented in Table 2. DEEP-ALIGN is significantly faster than Sinkhorn and Activation Matching while achieving comparable results. Furthermore, DEEP-ALIGN is on par with Weight Matching w.r.t runtime, yet it consistently generates better weight alignment solutions.

## F  ADDITIONAL EXPERIMENTAL RESULTS

**Scaling to large networks.** Here we show that DEEP-ALIGN scales well, and can be used to align widely used large architectures. Specifically, we use two datasets of VGG11 (9M parameters) and VGG16 (15M parameters) networks, trained on CIFAR10. Each dataset consists of 4500 training examples (similar to the CNN experiment in the main paper), 100 networks for validation, and 100 for testing. The results presented in Table 5 and Figure 6 show that (1) DEEP-ALIGN scales well and can process widely used large architectures, and (2) DEEP-ALIGN performs well without increasing the data size. This demonstrates that our method is suitable for processing contemporary deep architectures.

**DEEP-ALIGN as initialization.** As discussed in the main text, our approach can be used as initialization for optimization-based approaches, like the Sinkhorn re-basin. Here, provide extended results on using the output of DEEP-ALIGN as the initial value for the alignment problem. We evaluate two previously proposed methods, weight-matching (WM) (Ainsworth et al., 2022) and Sinkhorn re-basin (Peña et al., 2023). Initializing the Sinkhorn method significantly improves the performance under all evaluated datasets. In addition, using DEEP-ALIGN initialization greatly improves the convergence speed. Furthermore, DEEP-ALIGN improves the barrier results of the weight-matching

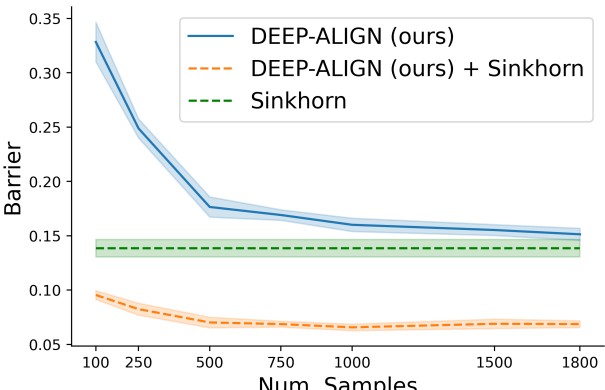

Figure 8: *Effect of sample size*: DEEP-ALIGN achieves on par results w.r.t the Sinkhorn with 1800 training pairs, while only 100 pairs are sufficient to significantly improve Sinkhorn by initializing the alg. with the DEEP-ALIGN outputs.

Table 6: *Optimizing* DEEP-ALIGN *using different objectives*: Test Barrier results averaged over 3 random seeds.

| | MNIST MLP CLS | CIFAR10 MLP CLS | CIFAR10 INR |
|---|---|---|---|
| $\ell_{\text{supervised}} + \ell_{\text{LMC}}$ | $0.007 \pm 0.00$ | $\mathbf{0.070 \pm 0.00}$ | $\mathbf{0.063 \pm 0.00}$ |
| $\ell_{\text{supervised}} + \ell_{\text{alignment}}$ | $0.061 \pm 0.00$ | $0.343 \pm 0.00$ | $0.127 \pm 0.00$ |
| $\ell_{\text{supervised}} + \ell_{\text{LMC}} + \ell_{\text{alignment}}$ | $\mathbf{0.005 \pm 0.00}$ | $0.078 \pm 0.00$ | $0.087 \pm 0.00$ |

method. Notably, using the DEEP-ALIGN initialization achieves on-par or improved values for the WM objective.

**Effect of sample size.** We evaluate DEEP-ALIGN on the sine-wave INR experiment from the main text, using a varying number of training examples. The results are presented in Figure 8. DEEP-ALIGN achieves on par results w.r.t the Sinkhorn method with random initialization using 1800 training pairs. On the other hand, initializing the Sinkhorn method with the DEEP-ALIGN alignment shows significant improvement in the test barrier using only 100 training pairs. These results show the efficiency of DEEP-ALIGN both in producing model alignments or initializing optimization-based approaches.

**Ablation on the DEEP-ALIGN objective.** Here, we provide results for DEEP-ALIGN trained with different objectives. Recall that we introduced three objectives (losses) to train DEEP-ALIGN. The first is the supervised loss $\ell_{\text{supervised}}$ computed using a model and its permuted version. The second is $\ell_{\text{alignment}}$ which is the $L2$ loss between the aligned weight vectors, and the third is $\ell_{\text{LMC}}$ which evaluates the original task loss on the line segment between the aligned models. For this ablation study, we use the MNIST and CIFAR10 MLP classifiers along with the CIFAR10 INRs. The results are presented in Table 6. Using only the supervised and alignment loss generally achieves insufficient results in terms of the Barrier metric. Dropping the alignment loss and using the supervised and LMC losses appears to have a minimal impact on the results in the classifier experiments. However, interestingly, including the $\ell_{\text{alignment}}$ in the INR experiment seems to have a detrimental effect on the Barrier results, causing a significant drop in performance. This suggests the alignment loss and the barrier metric are not always well aligned. In these cases it is advised to drop the $\ell_{\text{alignment}}$ loss and optimize DEEP-ALIGN with the $\ell_{\text{supervised}}$ and $\ell_{\text{LMC}}$ losses.

**Visualization of predicted permutations.** We visualize the predicted permutation obtained using DEEP-ALIGN applied to three test sine-waves INRs. Each network pair consists of an INR and its permuted and noisy version. For clarity, we depict only the first permutation matrix, $P_1$. The rows of Figure 9 correspond to the three test INRs. The left column represents the output from the $F_{prod}$ layer, which then projected to the set of permutations using $F_{prod}$ (middle column). DEEP-ALIGN is able to perfectly predict the ground truth permutations (right column).

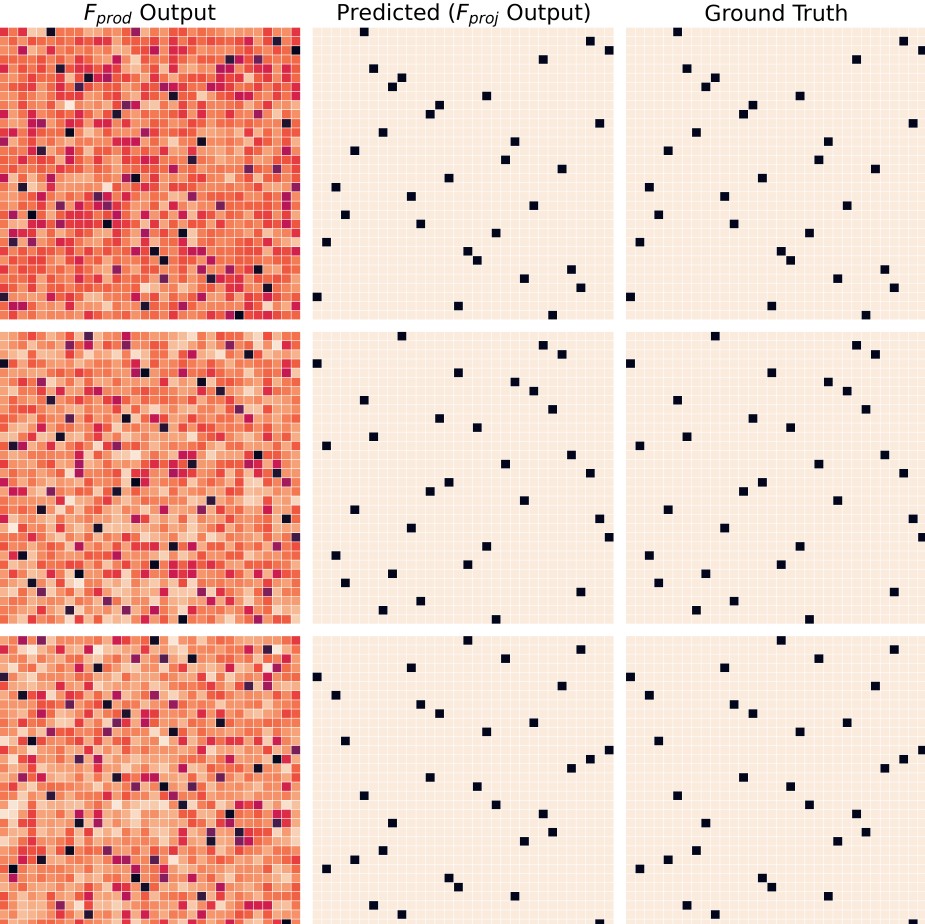

Figure 9: Predicted permutation matrices and ground truth permutations for three test sine wave INRs and their permutated and noisy version. DEEP-ALIGN outputs the exact ground truth permutations.

