# OpenReview forum: "Equivariant Deep Weight Space Alignment"
_ICLR.cc/2024/Conference — Submitted to ICLR 2024_

### Official Review · Reviewer_ayDH · 2023-10-20

**Soundness:** 3 good
**Presentation:** 3 good
**Contribution:** 3 good
**Rating:** 6
**Confidence:** 4

**Summary:**

This paper proposes a learning-based framework that learns to predict the optimal weight alignment between two neural networks. The proposed architecture is equivariance to permutation symmetry of the input neural networks. The authors prove that the approach can approximate the Activation Matching algorithm and guarantees to produce the correct alignment when a perfect alignment exists. At inference time, the framework aligns unseen network pairs without additional optimization. Experiment results show that the proposed approach is faster and produces better alignment than optimization-based approaches. The predicted alignment can also be used as initialization for optimization-based approaches to improve their alignment quality.

**Strengths:**

-	The idea of incorporating symmetries of the weight space is both natural and novel, and its effectiveness is clearly demonstrated in experiments.
-	An efficient method for finding weight alignment is an important problem in model merging, with potential applications in real-time merging in federated learning and weight-space clustering. It also allows more efficient experimentations in understanding the loss landscape of deep networks.
-	The paper is easy to follow. Writing is clear and concise.

**Weaknesses:**

-	The proposed framework requires new training for every new input neural network architecture, even for minor changes such as adding or modifying the size of a layer.
-	It is not clear how to choose the weights for the linear combination of the three loss functions in Section 4.2. Also, as shown in Appendix F, the two unsupervised loss functions are not always aligned, and it is not clear which loss to include before training. Finally, the evaluation metrics, barrier and AUC, are based on model merging tasks, but the supervised loss does not seem to help on these metrics directly.
-	The practical contribution would be more convincing if the authors can demonstrate the framework’s effectiveness in applications that require weight alignment.

### Minor:
-	$\rho_1$ and $\rho_2$ in the first paragraph of Section 2 are not defined. Are they representations?
-	Section 4.1: might be better to state what $\theta$ is somewhere.
-	Figure 5: The “Weight Matching” and “DEEP ALIGN + Weight Matching” have very similar colors.

**Questions:**

-	Does the exactness property (proposition 5) apply to test data as well?
-	How do the three losses interact with each other? Does the two unsupervised loss provably help improve the supervised loss?
-	Figure 3: Why does the loss decrease on the interpolation around $\lambda=$ 0.1 or 0.9, especially in CIFAR10 CNNs?
-	Why is incorporating equivariance beneficial to the model performance? It is intuitively clear that weight alignment methods should respect the symmetry of the neural networks, but there does not seem to be evidence of the link between incorporating equivariance and improved performance.

---

> ### Author Response · Authors · 2023-11-19
> **Response to ayDH**
>
> ***The proposed framework requires new training for every new input neural network architecture.***
>
> DEEP-ALIGN is not restricted to a single input network architecture and can be used to align a variety of input network architectures *using the same DEEP-ALIGN model*. This can be done by replacing the DWSNet encoder, with an architecture-agnostic model like the concurrent submissions of https://openreview.net/forum?id=ijK5hyxs0n  and https://openreview.net/forum?id=oO6FsMyDBt. There, the input neural networks are modeled as weighted graphs, and the weight space encoder is implemented using a graph neural network (GNN). That type of GNN architecture can be trained and applied to a wide range of neural architectures, with early experiments showing very promising generalization abilities to OOD architectures (see subsection 5.1 here https://openreview.net/forum?id=ijK5hyxs0n).
>
> ------------------
>
> ***How to select the weights for the linear combination of the three loss functions?; How do the three losses interact with each other?***
>
> Our framework enables the practitioner to adapt the loss to the task at hand.
> In our experiments, we observed that combining all losses with equal weighting works well for most learning setups, except in the INRs experiment where removing the L2 loss was beneficial.
>
> ----------------
>
> ***Demonstrate the framework’s effectiveness in applications that require weight alignment.***
>
> As the reviewer mentioned, one important application is Federated Learning (FL) where it was shown that alignment-based averaging, instead of the standard federated averaging, leads to better performance and reduced communication cost (see e.g., https://arxiv.org/pdf/2002.06440.pdf). Following the reviewers' comments and motivated by applications from FL, we added a new experiment in which we align models trained using disjoint datasets. To that end, we split the CIFAR10 dataset into two disjoint splits with different class distributions. We train our DEEP-ALIGN model to align CNN networks trained using the different datasets. The merged models achieve lower classification loss compared to the original models. The results are provided here and are added to the revised version of the paper (Subsection 6.1, page 9, “Aligning networks trained on disjoint datasets”).
>
> [*Link to results*](https://github.com/DeepAlignPaper/deep-align-results/blob/main/cifar10_disjoint_datasets.png)
>
> Other possible applications are alignment-based mixup as data augmentation for learning in deep weight spaces (see, e.g., http://arxiv.org/abs/2311.08851) and weight-space clustering, for example for analyzing large INR collections.
>
> --------------------
>
> ***Does the exactness property (proposition 5) apply to test data as well?***
>
> Yes. The result is very general and holds for all weights and all input vectors aside from a set of measure zero (which implies zero probability for any continuous distribution).
>
> ---------
>
> ***Why does the loss decrease on the interpolation around lambda=0.1 or 0.9?***
>
> This is a common phenomenon in model merging, as demonstrated in the git rebasin (https://arxiv.org/abs/2209.04836) and Sinkhorn matching (https://arxiv.org/abs/2212.12042) papers.
>
> -------
>
> ***Why is incorporating equivariance beneficial to the model's performance?***
>
> Equivariance is well known to be beneficial for learning data with symmetries (see, for instance, the geometric Deep Learning proto book https://arxiv.org/abs/2104.13478). Among the main benefits of this approach is that the model does not have to "memorize" different equivalent input representations of the same object, resulting in better generalization. Moreover, many equivariant architectures, such as the architecture used in this paper, have significantly fewer parameters and are considerably more efficient than fully connected networks. This is a result of the inherent parameter-sharing schemes that these equivariant models employ.
> We refer the reviewer to Figure 3 in the DWSNet paper, which compares equivariant and non-equivariant models. The performance gap as well as the improved sample complexity are substantial even for very simple tasks. We are happy to include a similar experiment for the weight alignment task if required.
>
> -----------
>
> ***Missing definition of rho_1 and rho_2; Similar colors in Figure 5.***
>
> Thank you. We fixed these issues in the revised version of the paper.

---

> > ### Comment · Reviewer_ayDH · 2023-11-23
> >
> > Thank you for your response. Replacing the encoder with a graph neural network is a promising way to make the proposed framework architecture-agnostic, but more empirical result is needed to back up this idea. Nevertheless, the proposed approach is clearly effective in solving the optimal weight alignment problem, so I will keep my score.

---

### Official Review · Reviewer_VRRk · 2023-10-30

**Soundness:** 4 excellent
**Presentation:** 3 good
**Contribution:** 3 good
**Rating:** 6
**Confidence:** 4

**Summary:**

The authors propose a data-driven method to perform weight matching between neural networks. It aims to solve the assignment problem by training a Siamese architecture to perform activation matching and produce permutation matrices. The architecture is trained on generated pairs of weight vectors, where one vector is a noisy, randomly permuted version of the other, as well as on unlabeled pairs.

The authors embed their method into a group theoretic framework and provide proofs for several properties of their architecture: it is equivariant to permutation and is exact, meaning that it, under mild assumptions, converges to an optimal solution if a zero error solution exists.

The method is evaluated on several weight matching benchmarks for image classification tasks and implicit neural field networks. It consistently outperforms previous work while being orders of magnitude faster.

**Strengths:**

- The presented method is grounded in theory, the authors provide useful theorems and the framework is technically sound
- The method applies existing concepts for data-driven assignment solvers to the weight matching problem, which is a novel contribution
- The method is well evaluated and results clearly outperform previous methods while being much faster
- The paper is mostly well-written and presented

**Weaknesses:**

- The network that maps from weight embedding to activation space and its connection to stage 1 is unclear (see below for questions).
- The networks generalization capabilities are limited. In order to outperform previous methods, the architecture has to be trained on different weights of the same network architecture, solving the same task as during inference. It still performs reasonably well on slight OOD weights though.

**Questions:**

- I don't fully understand the network mapping from weight embeddings to activation space. If the network just maps onto the bias vectors, the input weights do not have any influence on the estimated permutation anymore, would that be correct? This seems to be unintuitive to me and I would like the authors to clarify.
- Also, if my interpretation is correct, what is the point of feeding the weights $w$ to the network at all and not just use the bias? What is the purpose of the first stage of the architecture?

------------
I thank the authors for the provided clarifications - the method is clear to me now. In total, my score remains as it is.

---

> ### Author Response · Authors · 2023-11-19
> **Response to VRRk**
>
> ***The network's generalization capabilities are limited; The method has to be trained using the same network architecture, solving the same task as during inference.***
>
> As a learning-based method, our approach indeed requires training on the task of interest given data from the targeted data distribution. The upside, however, is that at inference time our learning-based approach results in high-quality alignments at a fraction of the computation time compared to non-learning-based approaches. This opens up the ability to use alignment in applications that require solving many alignment problems in real-time, such as federated learning and weight space clustering.
> Nevertheless, as the reviewer mentioned, our OOD performance is very reasonable and we believe it can be even further improved by incorporating recent GNN-based deep-weight space encoders that can process different architectures (such as these concurrent submissions: https://openreview.net/forum?id=ijK5hyxs0n  and https://openreview.net/forum?id=oO6FsMyDBt). That type of GNN architecture can be trained and applied to a wide range of neural architectures, with early experiments showing very promising generalization abilities to OOD architectures (see subsection 5.1 here https://openreview.net/forum?id=ijK5hyxs0n).
>
> -------------------
>
> ***Provide more explanations on mapping from weight to activation space; Why feed the weights W to the network at all and not just use the bias?***
>
> Thank you. We understand that this part was not adequately explained in the original paper. Following your comment and since weight space networks are new and relatively unknown, we expanded our previous work section to include more information about DWSNets in the revised version of the paper.
>
> In a nutshell, weight space networks are composed of weight space layers that map weights and biases $(W_1, b_1,...,W_M,b_M)$ to new weights and biases $(W'_1,b’_1,...,W'_M,b'_M)$.  Importantly, these layers combine information from both biases and weights to output the new set of weight and bias representations. Thus, the output biases, and consequently the predicted permutations are also a function of the input weights, and not only the input biases.
>
> The reason that using the output bias representation makes sense is that activations and biases reside in the same space. In fact, we use the terms bias space and activation space interchangeably in the paper, which we believe is what caused the confusion.

---

### Official Review · Reviewer_ZcWa · 2023-10-31

**Soundness:** 3 good
**Presentation:** 2 fair
**Contribution:** 2 fair
**Rating:** 6
**Confidence:** 3

**Summary:**

This work presents a learned algorithm to find permutations that align the weights of two networks (e.g. for federated learning, continual learning, model merging). The key innovation of the algorithm is it uses a weight space network to turn input weight vectors into embeddings which can then be aligned as in previous works (via distance in a metric space and Hungarian algorithm). This weight space network is trained using the Sinkhorn method to make the process of finding permutations differentiable.

**Strengths:**

This approach appears to improve upon and is complementary to existing approaches, in that it aims to give better permutations than weight matching and is faster than fully learned permutations (termed *Sinkhorn* in this work), but can also be combined with the *Sinkhorn* algorithm. Proofs of exactness and equivalence to activation matching guarantee that this method will be as similarly reliable as activation alignment, where the *Sinkhorn* algorithm does not have the same guarantees. The ability to change the objective function for training the deep weight network makes the algorithm more expressive, akin to *Sinkhorn* algorithm, e.g. so that it can directly optimize for linear connectivity rather than the $L_2$ norm between weights. Some of the results (e.g. table 2) look promising.

**Weaknesses:**

The main limitation I see is that results are not given for sufficiently large/complex models/tasks. For the barrier results, it is known (e.g. in Ainsworth et al. or Jordan et al.) that deeper networks are harder to align so that they are linearly connected, whereas easier networks (e.g. 3-layer MNIST networks) are relatively trivial to align. Also, it's not clear if Deep-Align reliably beats activation matching (which is still relatively fast and scalable), or if Deep-Align + Sinkhorn can beat weight matching + Sinkhorn.

The major methodological innovations could use much more explanation. The DWSNet would benefit from a full detailed description, given that it is central to the presented approach and based on a very recent work that may not be well known. The Siamese structure and why it guarantees equivariance to transposition should in particular be fully described.

**Questions:**

How does this method scale (re. accuracy and computation time) to larger networks (e.g. VGG models) and residual networks?
How does weight-matching + Sinkhorn compare with Deep-Align + Sinkhorn?

---

> ### Author Response · Authors · 2023-11-19
> **Response to ZcWa 1/2**
>
> ***How does DEEP-ALIGN scale to larger networks (e.g. VGG models)?***
>
> Thank you. Following this important comment, we added experiments with deeper architectures, VGG11 with 9M parameters and VGG16 with 15M parameters. We used the same amount of data, 4500 training examples, similar to the CNN experiments in the paper. The experiments show that: (1) DEEP-ALIGN scales well and can process large architectures; and (2) DEEP-ALIGN generalizes well when applied to larger networks without increasing the data size. This demonstrates that DEEP-ALIGN is suitable for processing standard deep architectures. The results are provided here and are added to the revised version of the paper (Appendix F, page 18, “Scaling to large networks”).
>
>
> |                       | CIFAR10 (VGG11) |                | CIFAR10 (VGG16) |                |         |
> |:---------------------:|:---------------:|:--------------:|:---------------:|:--------------:|:----------------------:|
> |                       |    Barrier ↓    |      AUC ↓     |    Barrier ↓    |      AUC ↓     |       Runtime (Sec) ↓                  |
> |         Naive         |    1.273±0.04   |   0.738±0.02   |    1.131±0.04   |   0.777±0.03   |           ---          |
> |        Sinkhorn       |    0.368±0.00   |   0.041±0.00   |    0.921±0.05   |   0.212±0.02   |         158.40         |
> |       DEEP-ALIGN      |    0.310±0.00   |   0.059±0.00   |    0.779±0.01   |   0.356±0.01   |          0.98          |
> | DEEP-ALIGN + Sinkhorn |  **0.099±0.01** | **0.000±0.00** |  **0.349±0.01** | **0.027±0.01** | 159.38 = 0.98 + 158.40 |
>
> [*Link to results*](https://github.com/DeepAlignPaper/deep-align-results/blob/main/cifar10_VGG.png)
>
> ----------------
>
> ***How does weight-matching + Sinkhorn compare with Deep-Align + Sinkhorn? Does Deep-Align reliably beats activation matching?***
>
> **Comparison to Weight matching + Sinkhorn**: Following your comment and feedback, we added comparisons to the weight-matching (WM) + Sinkhorn baseline in Tables 1 and 2, and Figure 3 in the revised version of the paper. *Both DEEP-ALIGN and DEEP-ALIGN+Sinkhorn constantly outperform the WM + Sinkhorn baseline in all datasets and model architectures*. We provide here the main results:
>
> |                       |    MNIST (MLP)   |                  |   CIFAR10 (MLP)  |                  |
> |:---------------------:|:----------------:|:----------------:|:----------------:|:----------------:|
> |                       |     Barrier ↓    |       AUC ↓      |     Barrier ↓    |       AUC ↓      |
> |        Sinkhorn       |   0.027 ± 0.00   |   0.002 ± 0.00   |   0.183 ± 0.00   |   0.072 ± 0.00   |
> |     WM + Sinkhorn     |   0.012 ± 0.00   | **0.000 ± 0.00** |   0.137 ± 0.00   |   0.050 ± 0.00   |
> |       DEEP-ALIGN      |   0.005 ± 0.00   | **0.000 ± 0.00** |   0.078 ± 0.01   |   0.029 ± 0.00   |
> | DEEP-ALIGN + Sinkhorn | **0.000 ± 0.00** | **0.000 ± 0.00** | **0.037 ± 0.00** | **0.004 ± 0.00** |
> |
>
> Results for CNNs:
>
> |                       |   CIFAR10 (CNN)  |                  |    STL10 (CNN)   |                  |                      |
> |:---------------------:|:----------------:|:----------------:|:----------------:|:----------------:|:--------------------:|
> |                       |     Barrier ↓    |       AUC ↓      |     Barrier ↓    |       AUC ↓      |    Runtime (Sec) ↓   |
> |        Sinkhorn       |   0.313 ± 0.01   | **0.000 ± 0.00** |   0.366 ± 0.00   |   0.163 ± 0.000  |         79.81        |
> |     WM + Sinkhorn     |   0.333 ± 0.01   | **0.000 ± 0.00** |   0.371 ± 0.00   |   0.165 ± 0.00   | 80.02 = 0.21 + 79.81 |
> |       DEEP-ALIGN      |   0.237 ± 0.01   | **0.000 ± 0.00** |   0.382 ± 0.01   |   0.182 ± 0.00   |         0.44         |
> | DEEP-ALIGN + Sinkhorn | **0.081 ± 0.00** | **0.000 ± 0.00** | **0.232 ± 0.00** | **0.097 ± 0.00** | 80.25 = 0.44 + 79.81 |
> |
>
> **Discussion on results w.r.t. activation matching**: DEEP-ALIGN outperforms activation matching in $3/4$ of the datasets and performs comparably on the remaining dataset. This is achieved while being almost $\times 20$ faster at inference time.

---

> > ### Author Response · Authors · 2023-11-19
> > **Response to ZcWa 2/2**
> >
> > Q: ”The major methodological innovations could use much more explanation”.
> > A:  Thank you for this important comment. Weight space networks are indeed new and relatively unknown. Thus, we expanded our previous work section to include more information about DWSnets in the revised version of the paper. In addition, we will attempt to explain the principle behind the equivariance to transposition. In short, this equivariance property means that when the input networks are swapped, i.e. we would like to map from $v_2$ to $v_1$ and not from $v_1$ to $v_2$, the output permutations should map in the opposite direction, and the mapping itself should be the inverse mapping. This is a fundamental principle in matching/alignment problems.
> > In our case, since these permutations are represented as permutation matrices, the inverse mapping is exactly the transposed permutation matrix. Using Siamese architecture is a natural way to implement this idea. For example, when we use a simple outer product layer as $F_{prod}$ we get the following for the $m$-th output:
> >
> > $F(v_1,v_2)_m = F_E (v_1)_m * F_E (v_2)_m^T$
> >
> > Where $F_E$ stands for DWS encoder. Note how changing the order of the inputs generates the transpose exactly as desired.
> >
> > $F(v_2,v_1)_m = F_E(v_2)_m*F_E(v_1)_m^T = F(v_1,v_2)_m^T$
> >
> >
> > Which means that we obtained the equivariant property we wanted.

---

> > ### Comment · Reviewer_ZcWa · 2023-11-20
> >
> > Thank you for the updated experiments which look promising. For a fair evaluation against the baseline methods, could you also include WM + Sinkhorn and activation matching in the evaluations on VGG-11 and VGG-16?

---

> > > ### Author Response · Authors · 2023-11-21
> > > **Response to ZcWa**
> > >
> > > Thank you for your valuable feedback.
> > > Certainly, we provide updated results for CIFAR10 (VGG11) experiment including Activation Matching and Weight Matching + Sinkhorn methods.  Due to time constraints, we will provide the full results for VGG16 (additional baselines) in the next revision of our paper.
> > >
> > > |                       | CIFAR10 (VGG11) |                | CIFAR10 (VGG16) |                |         |
> > > |:---------------------:|:---------------:|:--------------:|:---------------:|:--------------:|:----------------------:|
> > > |                       |    Barrier ↓    |      AUC ↓     |    Barrier ↓    |      AUC ↓     |       Runtime (Sec) ↓                  |
> > > |         Naive         |    1.273±0.04   |   0.738±0.02   |    1.131±0.04   |   0.777±0.03   |           ---          |
> > > |  Activation Matching | 0.576±0.01 | 0.204±0.00  |    ---                 |       ---              |           12.61
> > > |        Sinkhorn       |    0.368±0.00   |   0.041±0.00   |    0.921±0.05   |   0.212±0.02   |         158.40         |
> > > |  WM + Sinkhorn |    0.315±0.02  |     0.026±0.01   |    ---                 |       ---             |           171.10       |
> > > |       DEEP-ALIGN      |    0.310±0.00   |   0.059±0.00   |    0.779±0.01   |   0.356±0.01   |          0.98          |
> > > | DEEP-ALIGN + Sinkhorn |  **0.099±0.01** | **0.000±0.00** |  **0.349±0.01** | **0.027±0.01** | 159.38 = 0.98 + 158.40 |
> > >
> > > [*Link to updated results*](https://github.com/DeepAlignPaper/deep-align-results/blob/main/cifar10_vgg11_lmc.png)

---

> > > > ### Comment · Reviewer_ZcWa · 2023-11-23
> > > >
> > > > Hello, thank you for the quick experiment results. Given the positive comparisons I have raised my score. I do agree with the issues pointed out by other reviewers, such as how the method requires considerable training and how to scale the method to larger networks.

---

### Official Review · Reviewer_qGCa · 2023-11-01

**Soundness:** 3 good
**Presentation:** 3 good
**Contribution:** 2 fair
**Rating:** 6
**Confidence:** 2

**Summary:**

This paper attempts to solve the weight alignment problem by introducing a novel deep learning framework. The proposed DEEP-ALIGN method is fast and produce high quality alignment results after the pretrained process. DEEP-ALIGN does not require any labeled data but only depends on input network weight vectors. In addition to a theoretical analysis of the approach, experimental results support that a feed-forward pass with DEEP-ALIGN produces better or equivalent alignments compared to those produced by current optimization algorithms.

**Strengths:**

1. DEEP-ALIGN approach does not require labeled data but only the trained weight vectors.
2. DEEP-ALIGN performs on par or outperforms optimization-based approaches while significantly reducing the runtime or improving the quality of the alignments.
3. DEEP-ALIGN maintains good performance as an extra initialization step for Sinkhorn method on OOD data.
4. The theoretical analysis ensures the existence of the approximate solution that can be represented by DEEP-ALIGN architecture.

**Weaknesses:**

1. This method has to be pre-trained in advance. For experiments conducted on MNIST and CIFAR10, 8000 shallow networks are used to train the model to achieve the claimed performance. For deeper networks that are more common in nowadays deep learning task, this number may grow fast and the training cost can be unacceptable. Also, this paper does not provide experimental results about the relationship between the complexity of weight vectors and the scale of needed training data.

2. The real world applications for this method are unclear due to long the pre-train time. It seems that the traditional methods can compute the weight alignment much more effectively unless a ton of weight vectors need to be aligned so that the inference phase of DEEP-ALIGN dominates. What scenarios feature such characteristics?

**Questions:**

1. What is the definitions of $\rho_1$ and $\rho_2$ in the section 2 Preliminaries?
2. Is the assumption that the minimizer $k$ in equation (2) is unique reasonable? Any argument that such cases would be rare in practice?
3. I'm curious about the generalization capability of the model. As mentioned in the paper, the model will be first trained on a dataset of weight vectors and then applied to unseen weight vectors. Is there any possibility that one pertained DEEP-ALIGN model can work for networks of slightly different architectures? Due to the expensive cost of training phase, it can be helpful if one pertained model can work for more scenarios.

---

> ### Author Response · Authors · 2023-11-19
> **Response to qGCa 1/2**
>
> ***Is there any possibility that one pertained DEEP-ALIGN model can work for networks of slightly different architectures?***
>
> Yes. DEEP-ALIGN is not restricted to a single input network architecture and can be used to align a variety of input network architectures *using the same DEEP-ALIGN model*. This can be done by replacing the DWSNet encoder, with an architecture-agnostic model like the concurrent submissions of https://openreview.net/forum?id=ijK5hyxs0n  and https://openreview.net/forum?id=oO6FsMyDBt. There, the input neural networks are modeled as weighted graphs, and the weight space encoder is implemented using a graph neural network (GNN). That type of GNN architecture can be trained and applied to a wide range of neural architectures, with early experiments showing very promising generalization abilities to OOD architectures (see subsection 5.1 here https://openreview.net/forum?id=ijK5hyxs0n).
>
> -------------
>
> ***Scaling to deeper networks and the effect on the required amount of training data.***
>
> Thank you. Following this important comment, we added experiments with deeper architectures, VGG11 with 9M parameters and VGG16 with 15M parameters. We used the same amount of data, 4500 training examples, similar to the CNN experiments in the paper. The experiments show that: (1) DEEP-ALIGN scales well and can process large architectures; and (2) DEEP-ALIGN generalizes well when applied to larger networks without increasing the data size. This demonstrates that DEEP-ALIGN is suitable for processing standard deep architectures. The results are provided here and are added to the revised version of the paper (Appendix F, page 18, “Scaling to large networks”).
>
> |                       | CIFAR10 (VGG11) |                | CIFAR10 (VGG16) |                |          |
> |:---------------------:|:---------------:|:--------------:|:---------------:|:--------------:|:----------------------:|
> |                       |    Barrier ↓    |      AUC ↓     |    Barrier ↓    |      AUC ↓     |      Runtime (Sec) ↓                  |
> |         Naive         |    1.273±0.04   |   0.738±0.02   |    1.131±0.04   |   0.777±0.03   |           ---          |
> |        Sinkhorn       |    0.368±0.00   |   0.041±0.00   |    0.921±0.05   |   0.212±0.02   |         158.40         |
> |       DEEP-ALIGN      |    0.310±0.00   |   0.059±0.00   |    0.779±0.01   |   0.356±0.01   |          0.98          |
> | DEEP-ALIGN + Sinkhorn |  **0.099±0.01** | **0.000±0.00** |  **0.349±0.01** | **0.027±0.01** | 159.38 = 0.98 + 158.40 |
>
> [*Link to results*](https://github.com/DeepAlignPaper/deep-align-results/blob/main/cifar10_VGG.png)
>
> -------
>
> ***The real-world applications for this method are unclear due to long pre-train time.***
>
> Like other learning-based methods for solving combinatorial optimization problems, our method is particularly well-suited to applications that require solving many problems repeatedly.
> One key application is Federated Learning (FL). There, several models trained by clients need to be combined into a single model at the hub (model merging).
> This is often achieved by FedAvg, which merges models by averaging their weights elementwise in each round of communication. It has been shown (e.g., https://arxiv.org/pdf/2002.06440.pdf) that averaging aligned versions of these weights improves the performance of FedAvg, reduces the number of communication rounds, and improves training convergence.
>
> Motivated by applications from FL, we added a new experiment in which we align models trained using disjoint datasets. To that end, we split the CIFAR10 dataset into two disjoint splits with different class distributions. We train our DEEP-ALIGN model to align CNN networks trained using the different datasets. The merged models achieve lower classification loss compared to the original models. The results are provided here and are added to the revised version of the paper (Subsection 6.1, page 9, “Aligning networks trained on disjoint datasets”).
>
> [*Link to results*](https://github.com/DeepAlignPaper/deep-align-results/blob/main/cifar10_disjoint_datasets.png)
>
> Other possible applications are alignment-based mixup as data augmentation for learning in deep weight spaces (see, e.g., http://arxiv.org/abs/2311.08851) and weight-space clustering, for example for analyzing large INR collections.
>
> ------------
>
> ***What is the definition rho_1 and rho_2?***
>
> Thanks for spotting this. $\rho_1$ and $\rho_2$ are representations. We fixed the issue in the revised version of the paper.

---

> > ### Author Response · Authors · 2023-11-19
> > **Response to qGCa 2/2**
> >
> > ***Is the assumption that the minimizer k in equation (2) is unique reasonable? Any argument that such cases would be rare in practice?***
> >
> > Thank you for raising this question. Our equivariance results do not depend on the single minimizer assumption - this assumption was used to simplify the discussion. Equivariance holds even when multiple minimizers are present. This was stated in our submission (page 4, just after proposition 2), but we made efforts to make this clearer in the revised version of the paper.

---

### Author Response · Authors · 2023-11-19
**Rebuttal response**

We want to thank the reviewers for their time and effort, as well as their constructive and insightful feedback which we used to improve our manuscript. We are encouraged that reviewers found the paper novel (VRRk, ayDH), well-motivated (VRRk), efficient (ayDH), and easy to follow (ayDH). Notably, all the reviewers highlighted the theoretical contribution of this work (qGCa, ZcWa, VRRk, ayDH) with extended guarantees over previous works (ZcWa). Moreover, the reviewers found the experimental section showcased strong results, with our method outperforming existing approaches in accuracy and runtime (qGCa, ZcWa, VRRk, ayDH).

------

## Addressing Shared Concerns
We first address concerns raised by several reviewers:

***Add results for processing deeper networks (e.g. VGG models) (qGCa, ZcWa):***

Following this important comment, we added experiments with deeper architectures,   VGG11 with 9M parameters and VGG16 with 15M parameters. We used the same amount of data, 4500 training examples, similar to the CNN experiments in the paper.

The experiments show that: (1) DEEP-ALIGN scales well and can process large architectures; and (2) DEEP-ALIGN generalizes well when applied to larger networks without increasing the data size. This demonstrates that DEEP-ALIGN is suitable for processing standard deep architectures. The results are provided here and are added to the revised version of the paper (Appendix F, page 18, “Scaling to large networks”).

[*Link to figure*](https://github.com/DeepAlignPaper/deep-align-results/blob/main/cifar10_VGG.png)

|                       | CIFAR10 (VGG11) |                | CIFAR10 (VGG16) |                |          |
|:---------------------:|:---------------:|:--------------:|:---------------:|:--------------:|:----------------------:|
|                       |    Barrier ↓    |      AUC ↓     |    Barrier ↓    |      AUC ↓     |      Runtime (Sec) ↓                   |
|         Naive         |    1.273±0.04   |   0.738±0.02   |    1.131±0.04   |   0.777±0.03   |           ---          |
|        Sinkhorn       |    0.368±0.00   |   0.041±0.00   |    0.921±0.05   |   0.212±0.02   |         158.40         |
|       DEEP-ALIGN      |    0.310±0.00   |   0.059±0.00   |    0.779±0.01   |   0.356±0.01   |          0.98          |
| DEEP-ALIGN + Sinkhorn |  **0.099±0.01** | **0.000±0.00** |  **0.349±0.01** | **0.027±0.01** | 159.38 = 0.98 + 158.40 |

---------------

***Provide examples for applications of DEEP-ALIGN (qGCa, ayDH):***

Our method is particularly well-suited for applications that involve solving a large number of alignment problems. One such prominent application is Federated Learning (FL) where alignment-based averaging leads to better performance and reduced communication cost than the standard federated averaging (see e.g., https://arxiv.org/pdf/2002.06440.pdf). Motivated by applications from FL, **we added a new experiment** in which we align models trained using disjoint datasets. To that end, we split the CIFAR10 dataset into two disjoint splits with different class distributions. We train our DEEP-ALIGN model to align CNN networks trained using the different datasets. The merged models achieve better classification loss compared to the original models. The results are provided here and are added to the revised version of the paper (Subsection 6.1, page 9, “Aligning networks trained on disjoint datasets”).

[*Link to results*](https://github.com/DeepAlignPaper/deep-align-results/blob/main/cifar10_disjoint_datasets.png)

Other possible applications are alignment-based mixup as data augmentation for learning in deep weight spaces (see, e.g., http://arxiv.org/abs/2311.08851) and weight-space clustering, for example for analyzing large INR collections.

------------

***Can DEEP-ALIGN process varying network architectures (qGCa, VRRk, ayDH):***

Yes. DEEP-ALIGN is not restricted to a single input network architecture and can be used to align a variety of input network architectures using the same DEEP-ALIGN model. This can be done by replacing the DWSNet encoder with an architecture-agnostic weight-space encoders like the concurrent submissions of https://openreview.net/forum?id=ijK5hyxs0n  and https://openreview.net/forum?id=oO6FsMyDBt. There, the input neural networks are modeled as weighted graphs, and the weight space encoder is implemented using a graph neural network (GNN). That type of GNN architecture can be trained and applied to a wide range of neural architectures, with early experiments showing very promising generalization abilities to OOD architectures (see subsection 5.1 here https://openreview.net/forum?id=ijK5hyxs0n).

---

### Author Response · Authors · 2023-11-22
**To all reviewers**

We sincerely appreciate the thoughtful feedback from the reviewers. We carefully considered all comments and have uploaded a revised version that addresses the major concerns raised.  Notably, we added new experiments, showing that our approach outperforms all baselines on larger models as well.

If the reviewers feel we have adequately addressed their concerns, we would be grateful they would consider reassessing their rating.

We are happy to clarify or expand any part of the updated manuscript if helpful.

---

### Meta-Review · Area_Chair_zeN8 · 2023-12-11

**Metareview:**

The paper presents deep align a data driven method to perform alignment of weights between neural networks. Authors propose to learn a network that does this matching, the network takes as input a weight of a network and a noisy permuted version of it. Authors proved some theoretical claims about this method showing under some assumption that it leads to zero errors in the matching. The method is evaluated on  benchmarks for image classification tasks and implicit neural field networks.

While this method leads to some improvement in the matching, reviewers pointed out that  it has a costly pretraining of the matching network. It is not clear from the paper how to scale this method especially if the architecture weights are much higher dimensional and deeper. Hence despite its success on the networks considered it is not clear how to scale it to larger networks, nor how many networks it needs to train on, which is not feasible in this case.

 Authors justified this in the rebuttal that federated learning may need to resolve this alignment problem multiple times but did not provide timing experimentation w.rt to optimization baseline and the learning of the matching of the network to justify this amortization.

Overall the method proposed in the paper is natural but not expected to be impactful.

**Justification For Why Not Higher Score:**

The scalability of the method and the transfer learning of matching network to other architecture or datasets is not straightforward.

**Justification For Why Not Lower Score:**

N/A

---

### Decision · Program_Chairs · 2024-01-16

Reject